



# Variability in the mass absorption cross-section of black carbon (BC) aerosols is driven by BC internal mixing state at a central European background site (Melpitz, Germany) in winter

Jinfeng Yuan[1], Robin Lewis Modini[1], Marco Zanatta[2], Andreas B. Herber[2], Thomas Müller[3], Birgit Wehner[3], Laurent Poulain[3], Thomas Tuch[3], Urs Baltensperger[1] and Martin Gysel-Beer[1]

[1]Laboratory of Atmospheric Chemistry, Paul Scherrer Institute, Forschungsstrasse 111, 5232 Villigen PSI, Switzerland
[2]Helmholtz Center for Polar and Marine Research, Alfred-Wegener-Institute, Am Handelshafen 12, 27570 Bremerhaven, Germany
[3]Leibniz Institute for Tropospheric Research, Permoserstraße 15, 04318 Leipzig, Germany

*Corresponding author*: Robin Lewis Modini (robin.modini@psi.ch)

**Abstract.** Properties of atmospheric black carbon (BC) particles were characterized during a field experiment at a rural background site (Melpitz, Germany) in February 2017. BC absorption at a wavelength of 870 nm was measured by a photoacoustic extinctiometer and BC physical properties (BC mass concentration, core size distribution and coating thickness) were measured by a single-particle soot photometer (SP2). Additionally, a catalytic stripper was used to intermittently remove BC coatings by alternating between ambient and thermo-denuded conditions. From these data the mass absorption cross section of BC ($MAC_{BC}$) and its enhancement factor ($E_{MAC}$) were inferred. Two methods were applied independently to investigate the coating effect on $E_{MAC}$: a correlation method (ambient $MAC_{BC}$ vs BC coating thickness) and a denuding method ($MAC_{BC,amb}$ vs $MAC_{BC,denuded}$). Observed $E_{MAC}$ values varied from 1.0 to 1.6 (lower limit from denuding method) or ~1.2 to 1.9 (higher limit from correlation method) with the mean coating volume fraction ranging from 54 to 78 % in the dominating mass equivalent BC core diameter range of 200–220 nm. $MAC_{BC}$ and $E_{MAC}$ were strongly correlated with coating thickness of BC, while other factors were found to have a potential minor influence as well, including air mass origins (different BC sources), mixing morphology (ratio of inorganics to organics), BC core size distribution and absorption Ångström exponent (AAE). These results for ambient BC measured at Melpitz during winter show that the lensing effect caused by coatings on BC is the main driver of the variations in $MAC_{BC}$ and $E_{MAC}$, while changes in other BC particle properties such as source, BC core size or coating composition play only minor roles.

## 1 Introduction

Black carbon (BC), which commonly refers to graphitic elemental carbon (Petzold et al., 2013), is a major component of atmospheric aerosols. BC-containing particles are emitted from incomplete combustion processes. BC is highly refractory, insoluble and a strong light absorber across the whole solar spectrum (Corbin et al., 2019). The latter makes BC the dominant light-absorbing component of atmospheric aerosols and causes a substantial positive radiative forcing through aerosol-



radiation interactions (Bond et al., 2013). Two parameters are required to quantify the light absorption coefficient of BC ($b_{\mathrm{ap,BC}}$; [Mm$^{-1}$]) in climate models: the mass absorption cross section of black carbon (MAC$_{\mathrm{BC}}$; [m$^2$ g$^{-1}$]) and the BC mass concentration ($m_{\mathrm{BC}}$; [µg m$^{-3}$]), as shown in Eq. (1),

$$b_{\mathrm{ap,BC}} = \mathrm{MAC_{BC}} \times m_{\mathrm{BC}} , \tag{1}$$

BC mass concentrations are simulated with chemical transport models taking BC emission inventories as input. MAC$_{\mathrm{BC}}$ values must be assumed or calculated from simplified optical models based on knowledge gained from laboratory and field measurements.

The term "aerosol mixing state" is used to describe the morphology of a particulate species of interest, including externally
and internally mixed states. For example, when BC is freshly emitted, it is separated from other species, which refers to the externally mixed state. During the atmospheric aging of BC, non-BC species coagulate (e.g. particulate sulfate, nitrate, organics) with or condense (gas vapors) onto BC particles to form a variety of internal mixing states.

The mixing state of BC with other particulate matter in the same particle – the internal mixing state – is relevant because it
influences the light absorption by the BC in this particle. Based on a simple configuration of concentric spheres core-shell morphology, Mie theory provides a solution to predict light absorption by coated BC particles (Bohren and Huffman, 1998). According to this theory, if a BC particle is coated with non-absorbing species, more light is refracted towards the BC core, enhancing the amount of light it absorbs, and thereby increasing its MAC$_{\mathrm{BC}}$ value (Eq. 1). This is known as the "lensing effect" (van de Hulst, 1957). The MAC$_{\mathrm{BC}}$ enhancement factor, $E_{\mathrm{MAC}}$, due to lensing is conceptually defined as the MAC value of the
mixed particle, MAC$_{\mathrm{BC,mixed}}$, divided by the MAC value of the bare BC core:

$$E_{\mathrm{MAC}} = \frac{\mathrm{MAC_{BC,mixed}}}{\mathrm{MAC_{BC,bare}}} , \tag{2}$$

The lensing effect cannot result in unlimited enhancement of light absorption. Instead, a saturation point occurs, above which $E_{\mathrm{MAC}}$ does not increase any further with continued increase in coating thickness. The maximal MAC enhancement factor that can be reached for a particle depends on particle morphology and size, with greater values for smaller particles. Mie theory
(Bohren and Huffman, 1998) predicts that the lensing effect saturates at an enhancement factor of 2–3 for concentric coatings around BC cores with mass equivalent diameters in the range between 100 and 300 nm, which is the diameter range where ambient BC mass size distributions typically peak (Bond et al. (2006). Therefore, $E_{\mathrm{MAC}}$ factors of up to ~ 3 are plausible for ambient BC particles (Bond et al., 2006; Zhang et al., 2017).

Particle morphology, i.e. the shape of the particle and the core, as well as the position of the BC core within a particle, also affect absorption enhancement. More sophisticated numerical simulations of light absorption by BC particles of variable morphology indicate that the fractal dimension, the location inside the particle and the refractive index of BC aggregates also





influence $E_{MAC}$. Considering these factors typically results in smaller $E_{MAC}$ than would be expected with simpler core-shell morphologies (Adachi et al., 2010; Zhang et al., 2017).


As well as being theoretically predicted, the lensing effect has also been observed in laboratory experiments. For example Shiraiwa et al. (2010) coated colloidal graphite particles (compact, near-spherical shape) with volatile organic species (oleic acid and glycerol, boiling points are 290 °C and 360 °C, respectively) and measured their BC core size distribution, coating thickness and light absorption properties of both untreated and thermo-denuded samples (at 400 °C) . They observed that $E_{MAC}$

at 532 nm increased from 1.3 to 2 as the coating volume fraction was increased from 42% to 88%. In contrast, Qiu et al. (2012) observed a negligible lensing effect when they coated 150 nm combustion soot particles with secondary organic aerosol formed by the OH-initiated oxidation of toluene in an environmental chamber. The observed $E_{MAC}$ values at 532 nm only reached 1.1 for volume equivalent coating fractions from 54% to 70%.

The BC lensing effect and its relationship to internal mixing state have also been investigated with field measurements of atmospheric aerosols. Some studies have used a soot-particle aerosol mass spectrometer (SP-AMS, Aerodyne Inc.) to measure relative coating masses as an indicator for BC internal mixing state. The SP-AMS is a modified form of the aerosol mass spectrometer (AMS) performed by coupling a 1064 nm laser source to an AMS instrument, making it possible to measure both BC and non-refractory aerosol components. Using this instrument one can measure the chemical composition of BC-containing

particles (specifically, the average ratio of non-BC to BC core mass, referred to as $R_{coat-BC}$). However, the instrument is not quantitative in an absolute sense because the detection efficiency of BC cores in an SP-AMS depends on BC mixing state (Taylor et al., 2015) .

Other field studies have employed traditional single-particle soot photometer (SP2) instruments to measure absolute BC

coating thicknesses. Unlike the SP-AMS, the detection efficiency of BC particles in the SP2 does not depend on particle mixing state. Therefore, SP2 measurements are more quantitative than SP-AMS measurements in terms of both rBC mass and BC core diameter. In addition, incandescence measurements are combined with optical measurements of particle size in the SP2, allowing quantitative measurement of the BC coating thickness (Gao et al., 2007; Laborde et al., 2012a).

Cappa et al. (2019) summarized the most recent ambient observations of BC mixing state and the lensing effect. In some studies (Liu et al., 2015b; Peng et al., 2016) a strong lensing effect was observed, with $E_{MAC}$ reaching above 2 for mean ratios of coating to core mass ($R_{coat-BC}$) values up to 6. However, in other studies, only a weak or negligible lensing effect was observed ($E_{MAC} < 1.2$ for $R_{coat-BC}$ in the range 0.6 to 20) (Healy et al., 2015; Cappa et al., 2019). Cappa et al. (2019) formulated two hypotheses to explain the large difference in $E_{MAC}$ values observed in these two different groups of studies: 1) the diversity

of coating mass fraction among individual particles (e.g., Fierce et al., 2016); and 2) different mixing morphologies (e.g. off-center behavior of BC within a particle caused reduced $E_{MAC}$, Adachi et al., 2010). Furthermore, the composition of the coating





material may, via composition dependence of mixing morphology (Moffet et al., 2016), also affect the resulting lensing effect (e.g. Zhang et al., 2018; Wei et al., 2013). However, the relative importance and interplay of these effects in atmospheric aerosols remain poorly understood.


Zanatta et al. (2016) reported $MAC_{BC}$ values inferred from long-term observations at various European sites of the European Research Infrastructure for the observation of aerosol, clouds and trace gases (ACTRIS), additionally providing indirect evidence that the lensing effect occurs. In this study, we performed an intensive field experiment at one of these sites - the Melpitz observatory in Germany - with the goal to directly quantify the main drivers behind variations of $MAC_{BC}$, with a

particular focus on mixing state and lensing effect.

## 2 Method

### 2.1 Methods to quantify the lensing effect

To explore the light absorption and the coating induced enhancement by the same BC mass, the mass absorption cross section of BC ($MAC_{BC}$) is a key parameter to start with. Inverse to the modeling calculation shown in Eq. (1) in Sect. 1, the $MAC_{BC}$

([$m^2$ $g^{-1}$]) is defined in Eq. (3). To infer $MAC_{BC}$, the BC absorption coefficient ($b_{ap,BC}$; [$Mm^{-1}$]) at 870 nm and the BC mass concentration ($m_{BC}$; [$\mu g$ $m^{-3}$]) need to be measured.

$$MAC_{BC} = \frac{b_{ap,BC}}{m_{BC}},\qquad\qquad (3)$$

Two independent approaches were chosen in this study to quantify the lensing effect on the $MAC_{BC}$ of atmospheric BC-containing particles. The first approach, hereafter referred to as the correlation method, is based on a correlation analysis of concurrent quantitative measurements of both $MAC_{BC}$ and BC particle mixing state. The mixing state is inferred as coating

thickness resulting from the difference of measured diameters between the entire particle and the BC core based on an assumed core-shell configuration. The second approach, hereafter referred to as the denuding method, is based on modifying the mixing state of atmospheric BC-containing particles, i.e. to remove the coatings present on atmospheric BC using a catalytic stripper. Measurements of the $MAC_{BC}$ in both the untreated ambient aerosols and the corresponding denuded aerosols make it possible

to establish the causal relationship between the lensing effect and BC mixing state. For this purpose, the $E_{MAC}$ is calculated with Eq. (4) under the assumption that the $MAC_{BC}$ of the denuded aerosol represents the properties of bare BC.

$$E_{MAC} = \frac{MAC_{BC,ambient}}{MAC_{BC,denuded}},\qquad\qquad (4)$$

The two approaches described above were applied during a field experiment, described in details in the following.





## 2.2 The Melpitz site

The intensive field campaign was conducted at the research site of the Leibniz Institute for Tropospheric Research (TROPOS) in Melpitz (12°56′E, 51°32′N, 86 m a.s.l.). The Melpitz site is a rural and regional background site, belonging to many international (GAW, ACTRIS and EMEP) and domestic (GUAN) observational networks (Birmili et al., 2016; Poulain et al., 2014; Spindler et al., 2010; Spindler et al., 2013). The station is located near the town of Torgau with 20 000 inhabitants in eastern Germany and 50 km northeast from the city of Leipzig, with 600 000 inhabitants. The observational containers are

situated on the flat and semi-nature meadow surround by agricultural land. A federal main road (B 87) is 1.5 km north from the station and two conservative forests are located 2.5 km and 1 km in the north and south direction, respectively. The Melpitz site is about 130 km from the Polish border and anthropogenic emissions between Melpitz and Poland are negligible (Spindler et al., 2013). The measurements here are regarded as representative of the lowland background atmosphere in central Europe (Asmi et al., 2011; Aas et al., 2012; Birmili et al., 2009). The two main wind directions for Melpitz are South-West (SW) and

East (E) with the different air masses arriving at Melpitz: air masses crossing the western part of central Europe including the city of Leipzig and continental air masses with anthropogenic emissions from countries in the east of Europe via long-range transport. Seasonally, the particulate mass concentration is highest with the East wind direction in winter and lowest with the West wind direction in summer (Spindler et al., 2013).

The field experiment was conducted from 01 to 23 February in 2017. Contributions from different BC sources are expected during the winter season and the higher pollution level makes it possible to achieve higher time resolution with the online instruments.

## 2.3 Experimental setup

The sampling set-up including the instruments is shown in Fig. 1. The ambient air was passed through a $PM_{10}$ inlet followed

by a Nafion dryer (RH<30%) and a flow splitter. A first branch fed the instruments permanently probing untreated ambient aerosol, including a multi-angle absorption photometer; MAAP (aerosol absorption coefficient at 637 nm wavelength), an Aethalometer AE33 (spectral dependence of the aerosol absorption coefficient), and an aerosol chemical speciation monitor, ACSM. The ACSM (Aerodyne Research, MA, US; Ng et al., 2011) measured the near-PM1 bulk chemical composition of the non-refractory aerosol components (the instrument inlet has an upper cut-off at an aerodynamic diameter of around 1 µm).

A second branch directed sample to a 3-way valve, from which the aerosol passed through either a catalytic stripper or a bypass line. The valve was automatically switched every 15 minutes alternately delivering untreated ambient or denuded aerosol to a photo-acoustic extinctiometer (PAX) and a single-particle soot photometer, which measured the aerosol absorption coefficient at 870 nm wavelength and BC properties, respectively. The measurements were averaged to 3 h intervals (separately for each branch behind the switching inlet), as the signal-to-noise ratio of the 15 min data was insufficient for some instruments.



Beside the aerosol measurements, co-located trace gas measurements (including $SO_2$, NO and $NO_2$) were also performed (not shown in Fig. 1), which were used to indicate air mass origins combined with the patterns of meteorology and aerosol chemical properties (Sect. 3.1).

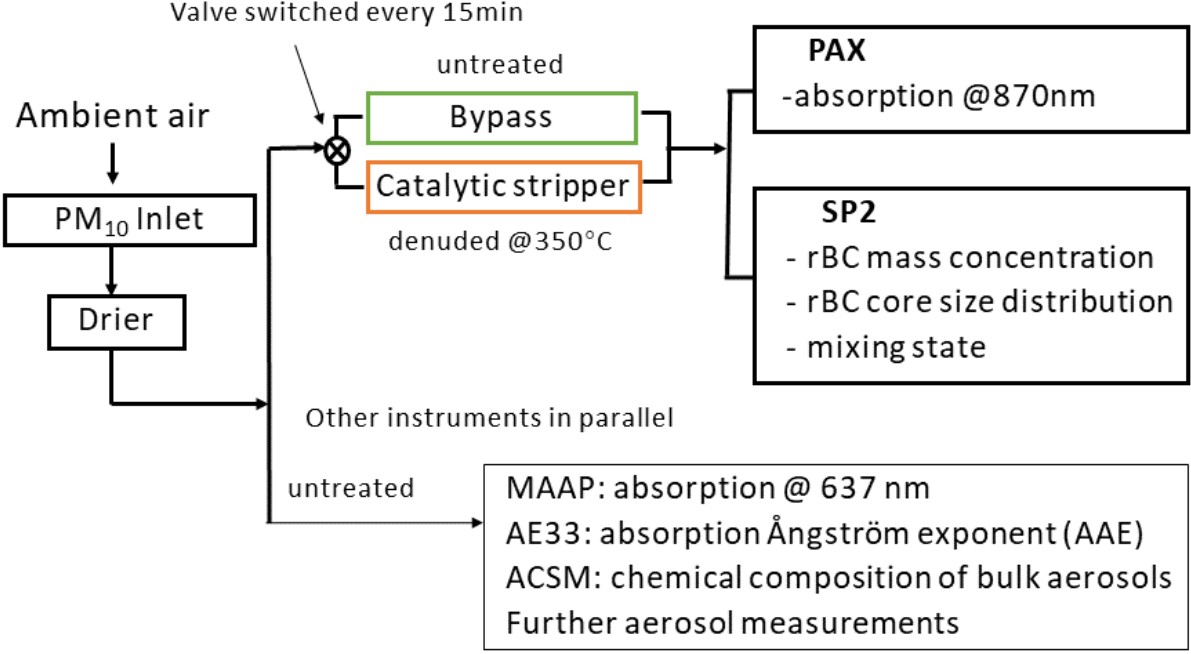


**Figure 1. Schematic of the sampling system, instruments and target quantities related to this study. PAX is photoacoustic extinctiometer and SP2 is single particle soot photometer. PAX and SP2 analyzed both ambient and denuded samples. The other instruments only analyzed the untreated ambient aerosol, including a multi-angle absorption photometer (MAAP), an Aethalometer (AE33), an aerosol chemical speciation monitor (ACSM) and further instruments, which are not shown here.**


## 2.4 Instrumentation

### 2.4.1 The single particle soot photometer (SP2)

The SP2 is based on the laser induced incandescent technique (LII) and the principles have been described in previous studies (Stephens et al., 2003; Schwarz et al., 2006). Briefly, particles pass through an intra-cavity, continuous-wave laser beam (Nd: YAG at 1064 nm). Since BC is the most strongly light absorbing and refractory aerosol material in the atmosphere (Pöschl, 2003; Schwarz et al., 2006), BC can absorb infrared light (Schwarz et al., 2010) and be heated to its boiling point as high as around 4230-4325 K (Moteki and Kondo, 2010), emitting thermal radiation as measurable incandescence before sublimation. The emitted incandescent light is optically filtered before being detected by a photomultiplier tube (PMT), which is sensitive




to a wide wavelength range (broadband between ~ 350 and 800 nm), equipped with high gain and low gain. The peak intensity of the incandescent light is proportional to the BC mass in the particle regardless of its mixing state (Slowik et al., 2007) since non-BC species evaporate at temperatures much lower than the BC sublimation temperature. Based on this operational characteristic, BC quantified by the LII method is termed as refractory BC (rBC) hereafter, following the terminology recommended by Petzold et al. (2013).


The inherent limitation for detection of BC by the SP2 is that for very small rBC mass in a single particle or insufficient lower laser intensities the conductive cooling dominates over BC absorption and thus BC cannot be heated to sublimation (Schwarz et al., 2010). The SP2 used in this study has 100% number-detection efficiency on a single particle basis when the rBC mass per particle is above ~0.5 fg, which corresponds to an rBC mass equivalent diameter, $D_{rBC}$, of ~80 nm using a void-free material
density of 1.8 g cm$^{-3}$. For BC cores greater than the upper limit of detection of the SP2, the particles are detected while the rBC mass cannot be quantified due to the signal saturation from the A/D converter. Therefore, the upper quantification limit (LOQ) is ~200 fg rBC mass per particle, corresponding to an rBC mass equivalent core diameter ($D_{rBC}$) of ~600 nm. Note that the SP2 is also equipped with a narrow band incandescence detector (NB: 630~800 nm; Schwarz et al., 2006). However, we did not further use these signals as the upper LOQ was at similar BC core mass as for the broadband detectors, and because
two-color pyrometry is out of scope of this manuscript. The BC mass of cores outside the size detection limits of the SP2 was corrected by using a lognormal fit of the mass size distribution. For the campaign averaged rBC mass size distribution, the missing mass was 2.5 ± 1.4 % below the lower LOQ and 1.1 ± 0.4 % above the upper LOQ. The details of the missing mass correction are described in a separate paper (Pileci et al., 2020, in preparation).

The recommended calibration of the incandescence channels with an aerosol particle mass analyzer (APM) is described elsewhere (Laborde et al., 2012a; Baumgardner et al., 2012; Moteki and Kondo, 2010). Briefly, an empirical calibration of the relationship between BC mass in a particle and resulting incandescence signal amplitude is required. Mass selected fullerene soot from Alfa Aesar (stock #40971, lot #FS12S011) was shown to provide calibration curves that match the response of the SP2 to BC from diesel exhaust (Laborde et al., 2012a) and atmospheric BC (Moteki and Kondo, 2010). In this study we used
a different batch of fullerene soot (Alfa Aesar; stock #40971, lot W08A039) for SP2 calibration. The SP2 response to this batch was later shown to be equal to that of the batch FS12S011, and therefore this calibration is in full agreement with the recommended approach. The reproducibility of rBC mass using this calibration approach is estimated to be better than ±10% and to represent BC mass in atmospheric aerosols from different sources within ±20%.

Particles passing the laser beam also elastically scatter laser light, which is detected by avalanche photodetectors (APD). BC-free non-absorbing particles do not evaporate within the laser beam and remain unperturbed, and the scattering signal is determined by a standard APD. The amplitude of the scattering signal is proportional to the partial scattering cross section of the detected particle for the solid angle covered by the detector optics. From the measured scattering cross section the optical





diameter of the particles ($D_{opt}$) is inferred using Mie theory (Bohren and Huffman, 1998, i.e. assuming spherical particle shape).
The refractive index of the particles is assumed to be 1.50+0i, which typically provides optical diameters that agree on average
within a few percent with the mobility diameter of the particle (Fig. S1). Unit detection efficiency for the standard optical
sizing was achieved for optical diameters $D_{opt} > \sim$150 nm and the upper LOQ, which is restricted by detector saturation, was
at $D_{opt}$ = 500 nm. Absolute calibration of the scattering cross section measurement was done using spherical polystyrene latex
(PSL) size standards of 269 nm (Thermo Scientific) and calculation based on Mie theory with known scattering cross section.

The BC containing particles are heated to above 4000 K due to BC core absorption, such that the non-refractory coating
materials evaporate within the laser beam. Therefore, the measured scattering amplitude is not proportional to the scattering
cross section of the unperturbed particle. However, the leading edge of the light scattering signal still contains information on
the scattering cross section of the unperturbed particle. Interpretation of the leading edge signal is only possible if the laser
intensity profile is known and if the time axis of the scattering signal can be related to the position of the particle in the laser
beam. This is achieved with an additional position sensitive detector (PSD) introduced by Gao et al. (2007). Knowing the
particle position in the laser beam makes it possible to infer the optical diameter of the unperturbed particle before evaporation
onset in the leading edge of the laser beam, commonly referred to as leading-edge-only (LEO) method. Details on the LEO
optical sizing approach can be found in previous studies (Gao et al., 2007; Laborde et al., 2012a; Taylor et al., 2015).

Accurate optical sizing with the LEO method requires several validations. A first one is to verify that the position dependent
laser intensity is correctly accounted for. This is done by comparing the LEO results from the scattering detector with the
corresponding standard optical sizing for BC-free particles (see Fig. S2 (a) for details). The LEO sizing can also be done with
the PSD signal, in which case an adjustable factor is used to make the LEO results match the results of a normal scattering
detector, as shown in Fig. S2 (b). This approach implicitly ties the PSD detector to the calibration of the normal scattering
detector. We used the PSD detector instead of the normal scattering detector for the LEO sizing because the 10 % and 90 %
percentiles are much narrower for the former, as shown in Fig. S2 c and d.

The principle behind the LEO sizing can be applied to the time-resolved signal of the normal scattering signal at any position
in the laser beam (Laborde et al., 2012a). The scattering signal of a BC-containing particle at incandescence onset represents
that of the bare BC core, which makes it possible to infer its optical diameter ($D_{opt,BC}$). This is done with assuming spherical
shape and a certain BC refractive index ($RI_{BC}$). Figure S3 shows a comparison of the optical diameter of the BC core with the
BC mass equivalent diameter inferred from the incandescence signal. The refractive index of the BC core was chosen to be
1.75+0.43i, which made the two diameters match in this study. This value is lower than the value most commonly used for
ambient soot in other studies with SP2 measurement at various sites ($RI_{BC}$ = 2.26+1.26i; Moteki et al., 2010; Laborde et al.,
2012b; Laborde et al., 2013; Zanatta et al., 2018; Dahlkötter et al., 2014). The reason for achieving the BC diameter match
with a lower than usual refractive index is not known, nor should this approach be interpreted as an accurate refractive index





measurement. However, choosing it in this manner ensures bias-free measurements of the coating thickness of uncoated bare BC particles.


The optical size of the individual BC-containing particle ($D_{\text{total}}$) can be inferred from the combination of Mie model calculation, the measured scattering signal of the entire particle and the core diameter ($D_{\text{rBC}}$) (Taylor et al., 2015; Schwarz et al., 2008). Briefly, based on a concentric core-shell configuration, the Mie model is able to calculate the scattering cross-section of the BC core with the input of $D_{\text{rBC}}$ and $RI_{\text{BC}}$. Then $D_{\text{total}}$ can be inferred by the Mie model with the input of the refractive index of

the coating ($RI_{\text{coat}}$) with scattering by coating (difference between measured scatter signal of entire particle and Mie calculated BC core scattering signal). The particle scattering cross-sections with fixed $D_{\text{rBC}}$, $D_{\text{total}}$, $RI_{\text{BC}}$ and $RI_{\text{coat}}$ inputs by the Mie model are stored in a series of 2-D lookup tables for data analysis in the PSI SP2 toolkit run with Igor Pro (Wavemetrics, OR, USA). Finally, the coating thickness ($T_{\text{coat}}$) can be calculated with the reconstructed $D_{\text{total}}$ and $D_{\text{rBC}}$:

$$T_{\text{coat}} = (D_{\text{total}} - D_{\text{rBC}})/2 \,, \tag{5}$$

The coating thickness is more sensitive to $RI_{\text{core}}$ than $RI_{\text{coat}}$, which is consistent with previous sensitivity evaluation (Taylor et al., 2015). The precision of the coating thickness retrieved by the LEO method was estimated to be about ±20 % (Laborde et al., 2012b) for $D_{\text{rBC}}$ ranging from 150 to 400 nm.

### 2.4.2 Absorption measurements

### 2.4.2.1 Instruments

A MAAP (ThermoFisher Scientific, MA, USA; Petzold and Schönlinner, 2004) was used to measure the aerosol absorption coefficient at 637 nm. The MAAP measures both the light radiation transmitted and back scattered from a particle-loaded fiber filter, and determines the fraction of light absorption by absorbing aerosol components via a radiative transfer program. To minimize the interference by the light scattering aerosol components on the angular distribution of the back scattered radiation, the measurements are performed with three detectors at different angles. For the data analysis in this study, a factor of 1.05

has been applied for the required wavelength correction (from 660 to 637 nm) according to Müller et al. (2011).

A PAX (Droplet Measurement Technologies, CO, USA) was used to measure the aerosol absorption coefficient at 870 nm, which minimizes interference from light absorbing particulate matter other than BC such as brown carbon or dust. It applies photo-acoustic spectroscopy, which is described in detail in Arnott et al. (1999). Photo-acoustic spectroscopy has been widely

used in recent years as it is an in-situ measurement without perturbing particle morphology (Lack et al., 2006). The PAX was calibrated following the manufacturer instructions. However, the precision of these calibrations was insufficient and the absorption coefficients measured with application of these calibration coefficients were inconsistent with the results from the MAAP, as further discussed in Sect. 2.4.2.2 (Müller et al., 2011).



An Aethalometer (Model AE33, Magee Scientific, CA, USA; Hansen et al., 1984) was also used to measure the aerosol absorption coefficients at seven wavelengths (370, 470, 520, 590, 660, 880, and 950 nm). The AE33 is also a filter-based instrument, with associated limitations in quantifying the absorption coefficient absolutely. However, in this study we only applied the AE33 to infer the relative spectral dependence of light absorption. For this purpose we used the default instrument output, which includes a loading compensation and a correction for the effects of multiple scattering within the filter matrix

(Drinovec et al., 2015). Note that the absolute value of the multiple scattering correction is irrelevant for our purpose, except for the fact that it is assumed to be independent of wavelength.

The spectral dependence of the aerosol absorption coefficient, $b_{ap}$, often follows a power law $b_{ap}(\lambda) \sim \lambda^{-AAE}$ in good approximation. The exponent AAE is commonly referred to as the absorption Ångström exponent. In this study we use

absorption coefficients measured by the AE33 at two different wavelengths ($\lambda_1$ and $\lambda_2$) to infer the AAE using the following equation (Moosmüller et al., 2011):

$$AAE(\lambda_1, \lambda_2) = \frac{-\ln(b_{ap}(\lambda_1)/b_{ap}(\lambda_2))}{\ln(\lambda_1/\lambda_2)} \ , \tag{6}$$

BC absorbs light broadly from near-UV to near-infrared wavelengths and thus has a weak spectral dependence (AAE ≈ 1; van de Hulst, 1957). By contrast, light absorbing organics (Corbin et al., 2019), can exhibit substantial light absorption at near-UV

and blue wavelengths while being negligible at red to near-UV wavelengths, which normally results in AAE larger than 1.

Typical AAE values for traffic emissions are close to unity since BC is the only light absorbing component. However, AAE values are significantly larger than unity in biomass burning emissions due to co-emission of BC and brown carbon. The difference of the two branches of AAE values can be used for source apportionment via an "AAE model" (Zotter et al., 2017

Liu et al., 2014; Elser et al., 2016). Note that the application of the model is only precise under favorable conditions, in which traffic and biomass burning are the only sources.

### 2.4.2.2 Quantification of absorption measurement

As mentioned above, the PAX provides high time resolution in situ measurements. Moreover, the PAX measured both ambient

and denuded samples in this study. The MAAP provides absolute quantification since the differences among different instruments are less than 5 % (Müller et al., 2011). The AE33 provides wavelength dependence. In order to have quantitative absorption measurements, the absorption coefficient measured by the PAX ($b_{ap,PAX,870nm}$; [Mm$^{-1}$]) needs to be scaled to that of the MAAP ($b_{ap,MAAP,870nm}$; [Mm$^{-1}$]) at 870 nm. The original absorption coefficients measured by the MAAP were at 637 nm ($b_{ap,MAAP,637nm}$; [Mm$^{-1}$]). To compare $b_{ap}$ by the PAX and the MAAP at the same wavelength of 870 nm, $b_{ap,MAAP,637nm}$ values

were adjusted to that of 870 nm by the measured AAE values within the range of 637 to 870 nm as shown in Eq. (7):

$$b_{ap,MAAP,870nm} = b_{ap,MAAP,637nm} \times (637/870)^{AAE(637,870)} \ , \tag{7}$$


The correlation between the PAX measurements and the MAAP measurements scaled to 870 nm is excellent above ~1 Mm$^{-1}$ (Fig. 2). However, there is 31 % systematic bias between $b_{ap,PAX,870nm}$ and $b_{ap,MAAP,870nm}$, which is more likely an issue of inaccurate PAX calibration at high absorption coefficients. Since the MAAP is more robust as an absolute reference, the

$b_{ap,PAX,870nm}$ values were scaled by a factor of 1.44 to match $b_{ap,MAAP,870nm}$ as show in Fig. 2.

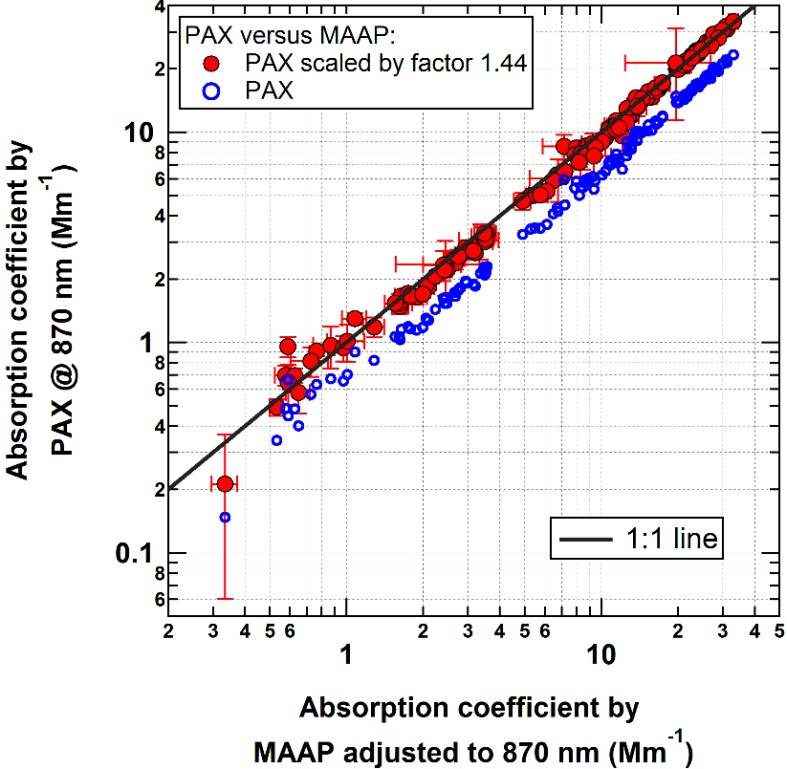

**Figure 2. Comparison of the absorption coefficients measured by the PAX and the MAAP. The MAAP data measured at 637 nm were adjusted to 870 nm using the spectral dependence of the absorption measured by the aethalometer. The PAX data measured**

**at 870 nm are shown with original calibration and after scaling by a factor 1.44 to match the MAAP data on average.**

**2.4.3 The Catalytic Stripper (CS)**

The basic principle and inner structure of the catalytic stripper (CS; Catalytic Instruments; Model CS015) has been described in Amanatidis et al. (2013). The residence time of the CS used in this study was approximately 0.35 s with the flow rate of 1.5

L min$^{-1}$, and the temperature was set to 350°C. As shown in Figure S5, the denuding process by the catalytic stripper did not influence the shape of the rBC core size distribution in the range ~60 to 600 nm. The fractional loss of rBC mass was up to 20% for $D_{rBC}$ below 300nm. The losses did not introduce any bias in the MAC$_{BC}$ values since the absorption measurement by



the PAX was also behind the CS. The median coating thickness was 58 and 32 nm before and after denuding, respectively, indicating that the CS did not remove the coating quantitatively.

## 3 Results and discussion

### 3.1 Periods with distinct air mass origin

The time series of wind direction and wind speed as well as chemical information are shown in Fig. 3. Three distinct periods and a short plume case were identified based on these data and on air mass back trajectory analyses (Fig. 4; calculated for an air mass arrival altitude of 100 m). The exact time windows and characteristics of the different periods are summarized in Table S1. In period 1 (02 Feb 2017 09:00 to 05 Feb 2017 21:00, UTC time), the local wind speed at Melpitz was low (median=1.2 m s$^{-1}$; IQR: 0.8 to 1.7 m s$^{-1}$) and the local wind direction at the site varied frequently. Back trajectory analysis showed that most of the air masses came from south to southwest (S to SW) of the sampling site, passing through the high Alps region. The local wind directions were generally not consistent with the air mass origin sector according to the back trajectory analysis. However, given the low and variable local winds during this time period, the back trajectory result is more relevant for interpreting the aerosol and gas phase composition. The gas phase mixing ratios were usually strongly dominated by NO$_x$ (except for period 2, see Table S1). The median SO$_2$ to NO$_x$ ratio was 0.08 (interquartile range IQR 0.05–0.11). Similarly, the aerosol properties were comparable among the periods except for period 2. For period 1, the median value of the total aerosol mass concentration from the integrated ACSM data (non-BC) and the SP2 (BC) was 10.6 (IQR 8.7–11.7) µg m$^{-3}$ (Table S1), the mass ratio of particulate inorganics to organics was 1.45 (IQR 1.22–1.55), and the sulfate to nitrate mass ratio was 0.60 (IQR 0.39–0.71). Note, these mass concentrations approximately correspond to PM1 composition due to the intrinsic upper detection limits of the ACSM and SP2. The aerosol composition measured in this study is consistent with previous observations from Melpitz and indicate that NO$_x$-rich vehicle emissions were a dominant source of pollution in these air masses (Spindler et al., 2013).





Figure 3. Time series of (a) wind direction and wind speed at 10 min time resolution measured 6 m above ground, (b) concentrations of gas phase species SO₂, NO, and NO₂ at 3 h time resolution, (c) mass concentrations of aerosol phase chemical components measured by the ACSM and the SP2 at 3 h time resolution, and (d) relative mass fractions of the measured aerosol chemical components.



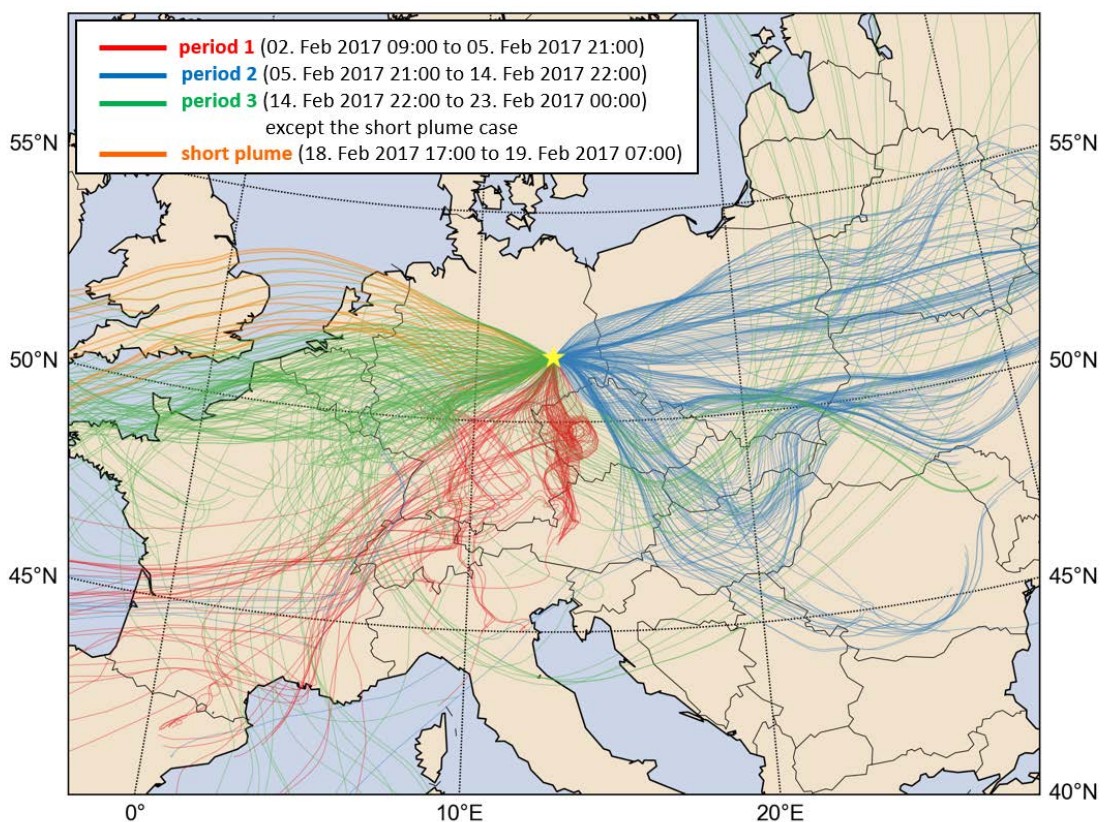

**Figure 4. Map of air mass back trajectories: 96-hour air mass back trajectories terminating at Melpitz site (yellow star) at an altitude of 100 m for every hour of the campaign. Trajectories terminating at altitudes of 10 m and 500 m were similar to those shown here. The map is based on the HYSPLIT atmospheric transport and dispersion modeling system provided by Air Resources Laboratory (ARL) (Stein et al., 2015).**

In period 2 (05 Feb 2017 21:00 to 14 Feb 2017 22:00), the local wind speed at the site (median 3.4, IQR 2.7–4.1 m s$^{-1}$) was higher than that of period 1 with constant local wind direction from the northeast to southeast (Fig. 3a). Back trajectory analysis indicated that the air masses arriving at the site had traveled from north-eastern, eastern and south-eastern Europe (Fig. 4), consistent with the local wind directions. In the gas phase, the most striking feature of the measurements is that SO$_2$ was present at significantly higher levels than in the other periods, with a median concentration of 9.2 (IQR 3.2–13.2) µg m$^{-3}$ (Fig. 3b). The SO$_2$ to NO$_x$ ratio was also much higher with a median value of 0.63 (IQR 0.28–0.83). The PM1 aerosol mass concentration from integrated ACSM (non-BC) and SP2 (BC) also showed the highest levels observed during the whole campaign period, with a median mass concentration of 23.0 (IQR 20.4–27.2) µg m$^{-3}$ (Table. S1), which was almost twice as high compared to the other periods. Despite the higher concentrations of chemical components (Fig. 3c), the relative composition of organic and total inorganic components (Fig. 3d) was similar to that of the period 1 and remained relatively stable within period 2. However, within the inorganic fraction, the sulfate to nitrate mass ratio was remarkably high with a



median value of 1.1 (0.9–1.2) during this period. The source of sulfur leading to the high concentrations of $SO_2$ and sulfate in
period 2 was likely residential and industrial solid fuel burning in east and south-east Europe, where coal with a high sulfur
content is still used as a fuel source. For example, Poland, located 150 km east of the Melpitz site, has the highest $SO_2$ emissions
among EU countries (Glasius et al., 2018). Previous studies have shown that combustion of coal and biomass (wood) are
significant sources of carbonaceous aerosols in Poland in winter (Spindler et al., 2013; Ciarelli et al., 2017; Glasius et al.,
2018).


In period 3 (14 Feb 2017 22:00 to 23 Feb 2017 00:00), the local wind at the site was dominated by westerly winds with
generally higher and more variable wind speeds (median 4 m s$^{-1}$, IQR 2.5–6.3 m s$^{-1}$) than those of period 1 and 2. The majority
of the back trajectories originate from western Europe (Fig. 4), consistent with local wind direction. In the particle phase, total
aerosol mass concentrations decreased steadily throughout period 3 to the lowest values observed during the whole campaign.
The median total aerosol mass concentration was 10.9 (IQR 8.1–15.8) µg m$^{-3}$ during the period (Table S1). Among the aerosol
components, substantially higher mass fractions of nitrate (35 %, 25–41 %) were found compared to the other periods, while
rBC mass concentrations (0.6, 0.3–1.4 µg m$^{-3}$) and rBC mass fractions (7%, 4–11%) reached their lowest levels for the whole
campaign (Fig. 3c and d, Table. S1). Organic mass fractions were less than 30% (Table. S1), lower than the fractions observed
in periods 1 and 2.


A short plume of BC aerosol passed over the sampling site between 18 Feb 17:00 and 19 Feb 07:00. Given the stagnant air
conditions and low wind speed during this period, this plume likely resulted from a local emission. As shown in Fig. 3c, the
3-hourly averaged rBC mass concentration peaked at ~ 4 µg m$^{-3}$ during this event, which was the highest value reached during
the campaign. The organic aerosol concentration was stable during the event (2.3, 2.1–2.5 µg m$^{-3}$), which suggests that the
plume did not come from a forest fire or biomass burning event since such events would emit large amounts of organics. A
coal burning source for the plume is also not evident since the $SO_2$ did not increase as shown in Fig. 3b. The fact that the
$AAE_{470\_950}$ dropped to around 1.0 during the period within the absorbing aerosol plume may indicate fresh emissions from a
combustion engine as BC source. In addition, the observed much larger BC core diameters (above 300 nm, Fig. 05b) in the
plume compared to the other periods likely indicated a super-polluter. Very dirty trucks or cars are known to produce larger
BC particles than typical engines do (Schneider et al., 2015); however, the persistence of the plume rather indicates a nearby
stationary rather than multiple mobile sources. In summary, the short plume seems to have resulted from a local event but the
exact source is not apparent.





## 3.2 Physical and optical properties of BC


**Figure 5.** Time series of the physical and optical parameters of BC at 3 h time resolution: (a) ambient rBC mass concentration measured by SP2 (left axis) and absorption coefficient of bulk aerosol measured by PAX at 870 nm (right axis), (b) model BC core
diameter of rBC mass size distributions determined with log-normal fits shown in Fig. S6 (left axis) and $AAE_{470\_950}$ (right axis), (c) mean thickness of non-BC coatings on ambient and denuded rBC cores with mass equivalent diameters in the range 200 to 220 nm, calculated from single particle data, (d) mean rBC volume fractions for ambient and denuded rBC particles calculated from the coating thickness data shown in (c), (e) mean $MAC_{BC}$ of ambient and denuded samples (left axis) and corresponding $E_{MAC}$ calculated with Eq. (4) (right axis). The gap in the denuded time series in (c) and (d) (and corresponding gap in the $E_{MAC}$ time series in (e))
from 06 to 14 Feb is due to malfunction of the valve switching system. The time periods when rBC mass was less than 0.2 µg m⁻³ are marked with a grey shading due to poor signal-to-noise ratio.





Time series of the physical and optical properties of BC are shown in Fig. 5. The BC mass concentration and absorption coefficient measurements were highly uncertain during periods of low aerosol loading (grey shaded periods occurring at the end of the campaign) due to the SP2 data acquisition settings (the SP2 was set to save data from only one out of every 200 particles) and instrumental limits (the lower quantification limit for the PAX during the campaign was ~1 Mm$^{-1}$, Fig. 2). Therefore, based on these considerations, a criterion of rBC mass concentrations $< 0.2\ \mu g\ m^{-3}$ was used to filter out data from further statistical analysis.

The ambient rBC mass concentration ranged between 1.00 and 3.03 $\mu g\ m^{-3}$ (IQR) throughout the whole campaign, but with systematic differences between the distinct campaign periods. Overall, the rBC mass concentration was higher in air masses from eastern Europe, consistent with previous observations at Melpitz in winter (Spindler et al., 2013). Specifically, the rBC mass concentration ranged from 1.79 to 3.48 $\mu g\ m^{-3}$ during period 2, increasing from ~ 0.5 to 4 $\mu g\ m^{-3}$ over the first four days of the period and then maintaining a relatively stable level at around 3.5 $\mu g\ m^{-3}$. In contrast, the rBC mass concentrations only ranged from 1.21 to 1.73 $\mu g\ m^{-1}$ during period 1 with air masses from southern and south-western Europe (Fig. 4). In period 3, the rBC mass concentration varied from 0.60 to 2.19 $\mu g\ m^{-3}$. At the beginning of this period, the rBC mass concentration decreased rapidly from ~4 to 1 $\mu g\ m^{-1}$ in a single day, as a result of the rapid switching of air arriving from eastern to western Europe. Following this sharp change, the rBC mass concentration was generally less than 1 $\mu g\ m^{-3}$ for the remainder of the period, which was likely the result of stronger dilution of emissions due to higher wind speeds and possibly also lower emissions in western Europe compared to eastern Europe.

The modal diameter of rBC mass equivalent diameter, calculated as the geometric mean of three-hourly averaged lognormal rBC mass size distributions ($D_{modal\_rBC}$), is shown in Fig. 5b. $D_{modal\_rBC}$ ranged from 186 to 240 nm throughout the whole campaign, which is a typical level for aged BC particles in continental remote or urban areas. For example, previous studies have observed $D_{modal\_rBC}$ values of ~240 nm in the European Arctic region (Zanatta et al., 2018), ~200 nm in continental air masses from eastern Europe measured in Paris (Laborde et al., 2013), and ~210-220 nm in Asian outflow measured at a remote site in Japan (Ueda et al., 2016). During this campaign, systematically larger median $D_{modal\_rBC}$ values were measured in period 2 (239 nm) than in periods 1 (190 nm) and 3 (181 nm). The larger BC particles measured in period 2 might be related to coal burning emissions (e.g. lignite coal burning in Poland): While the burning of hard coal briquette emits particles that lie mostly in the nuclei and Aitken mode (20-100 nm), the number size distribution of lignite emissions peaks in the accumulation mode (100-1000 nm) (Bond et al., 2002). Therefore, it is possible that BC cores from lignite burning are larger than BC from other common sources such as traffic. According to a summary in Kuchler and Bridge (2018), the production of lignite (60-70 million tons) has been almost the same as that of hard coal (70-80 million tons) in Poland from 2010 to 2015. During the short/distinct plume, $D_{modal\_rBC}$ ranged from 192 to 298 nm, substantially larger than the values measured in remote background air, which supports the interpretation that the plume resulted from local emissions.



### 3.3 BC mixing state, MAC$_{BC}$ and the lensing effect

The ambient MAC$_{BC}$ at 870 nm ranged from 7.2 to 7.9 m$^2$ g$^{-1}$ during the whole campaign, with a geometric mean value of 7.4 m$^2$ g$^{-1}$. These values are slightly higher than the MAC$_{BC}$ values calculated over 3 winter seasons in Melpitz and reported by

Zanatta et al. (2016). These authors applied a MAAP and thermal-optical elemental carbon mass measurements, and reported a MAC$_{BC}$ at 637 nm of 8.2 m$^2$ g$^{-1}$. This values corresponds to a MAC$_{BC}$ at 870 nm value of 5.3 to 5.7 m$^2$ g$^{-1}$, assuming AAE$_{637-870nm}$ values of 1.2 and 1.4, respectively. Nordmann et al. (2013) previously reported 7.4 m$^2$ g$^{-1}$ for MAC$_{BC}$ at 637 nm based on a week-long measurement combining MAAP and Raman spectroscopy, which is 10% lower than the value from Zanatta et al. (2016). The discrepancies between the values measured in this study and those reported by Zanatta et al. (2016) and Nordmann

et al. (2013) could be related to the fact that the studies were conducted over different time periods, or they might be due to the different techniques that were used to measure the BC mass concentration in each study. Pileci et al. (2020) showed from multiple field campaigns that co-located rBC and EC concentrations measurements can differ by as much as ~50 % in European background air. Given this fact, the agreement between the MAC$_{BC}$ values reported in this study and those previously reported is well within expectations.


The degree of internal mixing, expressed as the thickness of non-BC material coating the BC cores ('coating thickness'), was measured by the SP2 as described in Sect. 2.4.1. The modal diameters of the BC core mass distribution ($D_{modal\_rBC}$) were around 200–220 nm mass equivalent diameter during the campaign (Fig. 5b). Fortunately, these modal diameters were in the range of diameters for which the LEO-fit analysis described in Sect. 2.4.1 could be successfully applied to all types of internally mixed

BC particles (i.e., all types of BC particles ranging from uncoated to thickly coated particles). Therefore, to obtain a representative indicator of BC internal mixing state that is applicable to the mode of the BC mass size distributions, coating thicknesses are presented here for BC cores with diameters between 200–220 nm. For simplicity we hereafter to this parameter as 'coating thickness' without specifying the range of BC core diameters over which it has been calculated. The mean coating thickness at 3 h time resolution (Fig. 5a) was calculated from single particle data as shown in Fig. S7. The coating thickness

for ambient particles ranged from 51 to 61 nm (IQR) over the whole campaign, which corresponds to rBC volume fractions between 25 and 30 %.





**Figure 6. (a)** MAC$_{BC}$ **against the mean coating thickness of ambient rBC particles with core diameters in the range 200–220 nm mass equivalent diameter. (b)** MAC$_{BC}$ **of ambient and thermo-denuded samples against the mean coating thickness of ambient rBC particles. Each data point represents an average value over a 3-h time period with error bars representing the standard error of the mean. An orthogonal distance linear regression was applied to the ambient data to calculate a y-intercept value which represents an estimate of the MAC for zero coating, i.e., MAC$_{BC,bare}$. Note that the number of ambient MAC$_{BC}$ data points in (b) is smaller than in (a) due to the gap of missing data in the denuded MAC$_{BC}$ time series from 07 to 14 Feb shown in Fig. 5e.**





It is apparent from the time series in Fig. 5c and e that the ambient $MAC_{BC}$ and BC coating thickness correlated well. These two quantities are plotted as a scatter plot in Fig. 6a, indicating that ambient $MAC_{BC}$ and coating thickness were positively correlated with a Pearson correlation coefficient of 0.73. This provides evidence that there was indeed a lensing effect during

the campaign and that the BC mixing state was the main driver of $MAC_{BC}$ variability. We hereafter refer to this method of directly evaluating the dependence of $MAC_{BC}$ on the internal mixing state as the correlation method.

In order to estimate the $MAC_{BC}$ of bare, uncoated BC ($MAC_{BC,bare}$) a linear function was fit to the measurements (via orthogonal distance regression, ODR fit) to obtain a y-intercept of 5.0 m$^2$ g$^{-1}$. Since the relationship between $MAC_{BC}$ and coating thickness

may not be linear for lightly coated BC (the lensing effect appears to be weaker for lightly coated than moderately to heavy coated BC; Peng et al., 2016; Liu et al., 2017), this extrapolated intercept is regarded as a lower limit estimate of $MAC_{BC,bare}$.

A second, more direct approach was also applied to investigate the lensing effect – we refer to this as the denuding method. In this method a catalytic stripper (CS) was used to remove BC coating material (Fig. 5c) before the $MAC_{BC}$ measurement. Fig.

5e shows that this process resulted in lower $MAC_{BC}$ for the denuded samples relative to the corresponding ambient samples. In Fig. 6b, the ambient and denuded $MAC_{BC}$ values are plotted against the mean coating thickness of the unperturbed, ambient BC particles (i.e., before denuding by the CS). It is seen that the denuded-$MAC_{BC}$ values all fall in the range from 5.6 to 6.0 m$^2$ g$^{-1}$ (median= 5.8 m$^2$ g$^{-1}$) and that the values are largely independent of the original coating thickness, with a Pearson correlation coefficient of only 0.02. This is in contrast to the trend observed for the ambient $MAC_{BC}$ measurements, which

showed strong correlation with the coating thickness of the unperturbed BC particles. These results suggest that the coating material removed from the BC particles by the CS was largely responsible for the lensing effect displayed by the ambient BC particles.

The results displayed in Fig. 5c and Fig. S8 indicate that the CS did not remove all coating material from the BC particles,

likely due to the short residence time of around 0.35 s only. The denuded particles retain thin coatings which might still be responsible for a lensing effect. Therefore, we consider the median denuded $MAC_{BC}$ of 5.8 m$^2$ g$^{-1}$ to represent an upper limit estimate of $MAC_{BC,bare}$. The true value of $MAC_{BC,bare}$ likely falls within the range of 5.0 to 5.8 m$^2$ g$^{-1}$ defined by the lower and upper limit estimates arising from the correlation and denuding methods, respectively. Still, the results of the two methods are roughly consistent with each other, strengthening the conclusion that the internal mixing of BC drove the variability of $MAC_{BC}$

in this campaign.



## 3.4 Influence of other BC particle properties on MAC$_{BC}$

In this section, the importance of BC particle properties other than the internal mixing state is explored in relation to MAC$_{BC}$ variability.

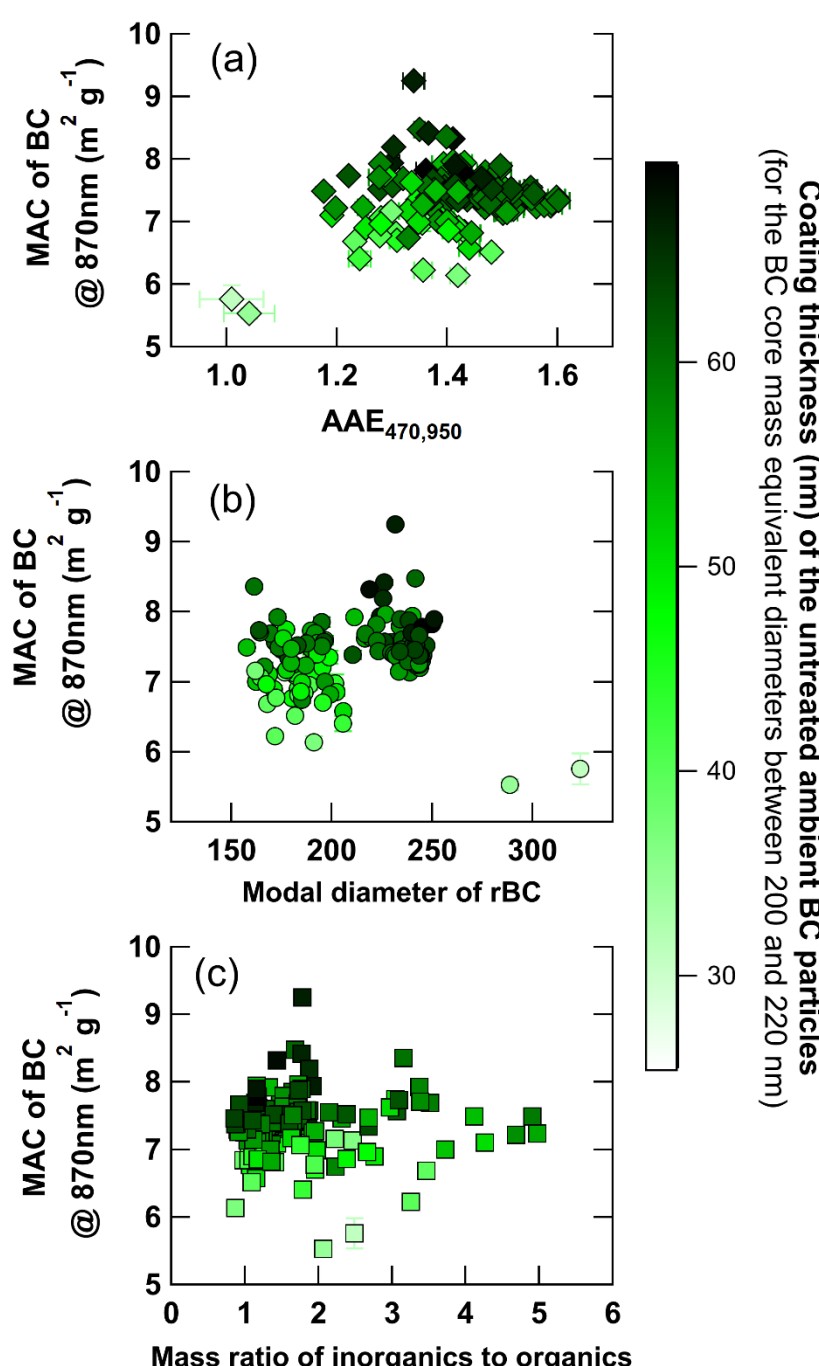





**Figure 7. Time-resolved MAC_BC values of untreated ambient BC particles at 870 nm wavelength calculated at 3 h resolution plotted against (a) the absorption Ångström exponent, AAE $_{470,950}$, between 470 and 950 nm wavelength, (b) the modal diameter of the BC core mass size distributions (D_rBC) (expressed as mass equivalent diameter), and (c) the ratio of non-refractory inorganic to organic components measured in near-PM1 by the ACSM.**


First, we address the question if the ambient MAC_BC is dependent on the BC "source". In a qualitative manner, the air mass origins characterized in Sect. 3.1 can be used to indicate different emission sources of BC particles in order to answer this question. In Fig. 6a, it is seen that the MAC_BC measurements from all air mass origin periods (Fig. 6a) scatter around the same regression fitting line. This suggests that the differences in MAC_BC between the periods are driven by differences in coating

thickness, rather than the air mass origins or the characteristics of BC from different sources.

As introduced in Sect. 2.4.2.2, the absorption Ångström exponent (AAE) can indicate different BC sources (e.g. traffic emissions typically display AAE ~1, while wood burning emissions generally have AAE >1). Therefore, in a more quantitative manner, the correlation of MAC_BC and AAE can be calculated to explore a possible source dependence for MAC_BC. Time

series of AAE values calculated between different pairs of wavelengths are shown in Fig. S9a, indicating similar behavior for all wavelength pairs. Given this fact, AAE$_{470,950}$ was chosen to explore the correlation with MAC_BC in Fig. 5b in order to have the wavelength dependence of absorption across a wide range of wavelengths (from blue to near-infrared).

AAE$_{470,950}$ ranged from approximately 1.2 to 1.6 during the campaign, except for the short plume period where values around

1 were observed. This indicates that there were contributions from emission sources other than traffic to BC during the campaign. Wood burning is performed in winter in central Europe for heating purposes, and it is likely that brown carbon emissions from this activity resulted in absorption at shorter wavelengths, contributing to high AAE$_{470,950}$ values (>1). If it is assumed that additional emission sources such as coal burning did not contribute to the sample, the aethalometer model for source apportionment (Zotter et al., 2017) can be used to separate the contributions of wood burning and traffic emissions to

total absorption (and therefore equivalent BC). The results of this model are shown in Fig. S9b. During periods 1 and 3, the model suggests that traffic and wood burning emissions contributed roughly equally to the observed total BC mass concentration. In contrast, during period 2, the modeled wood burning contribution dominated over the traffic contribution.

It is important to stress, however, that the AAE model can only apportion the measured absorption to two sources. As discussed

in Sec. 3.1, it is likely that a third source – coal burning emissions – also contributed to the BC measured during the campaign, at least in period 2 when the SO$_2$ to NOx ratio was significantly higher. Bond et al. (1999) observed that the industrial burning of lignite coal produced a yellowish, carbonaceous aerosol with strong absorption wavelength dependence. More generally, Bond et al. (2002) observed AAE$_{435,880}$ values between 1 and 3 for aerosol produced from the burning of different types of coal. Given the wide range of observed AAE for coal burning emissions, it is not possible to use measured AAE values to

apportion a specific fraction of equivalent BC to coal emissions. Still, the AAE$_{470,950}$ values found here were mostly above 1.4





during period 2, which is consistent with the assumption that coal burning (lignite) emissions were relevant during this period. Therefore, we conclude that coal burning emissions likely contributed to the BC measured during the campaign, however, without being able to quantify this contribution. In this case, the contributions of wood burning and traffic to BC as calculated with the aethalometer model and displayed in Fig. S9b should be considered as upper limit estimates.


Independent of a precise BC source apportionment the correlation of AAE with $MAC_{BC}$ can still be assessed to explore a potential source dependence for $MAC_{BC}$. Fig. 7a indicates there was no clear relationship between $MAC_{BC}$ and $AAE_{470,950}$ during the campaign. This supports the conclusion of the qualitative analysis displayed in Fig. 6a that $MAC_{BC}$ variability was driven by coating thickness, and not by the characteristics of BC from different sources.


Second, we address the question if the $MAC_{BC}$ depends on the BC core diameter. Figure 7b shows that there was no clear relationship between $MAC_{BC}$ and BC core diameter ($R^2=0.01$), which indicates that the variation in BC size was not responsible for the $MAC_{BC}$ variability. This is because the modal diameters of the BC core mass size distributions only varied within a relatively narrow range during the campaign ($D_{rBC}$ from 150 to 250 nm). This corresponds to dimensionless size parameters $x$

($= \pi D_{rBC} /\lambda$) in the range 0.5 to 0.9.  In both the Rayleigh (x<<1) and Mie regime (x~1), the size-distribution-weighted $MAC_{BC}$ is relatively independent of $D_{rBC}$. For much greater $D_{rBC}$ ($x \gg 1$, known as the geometric regime) incident light is unable to penetrate through the whole particle: absorption only occurs in the outer layer of the particle, which results in a strong negative relationship between $MAC_{BC}$ and $D_{rBC}$ in this regime. The modal diameters of the BC core mass size distributions observed during this campaign were not large enough to reach the geometric regime, which is why no clear relationship was observed

between $MAC_{BC}$ and $D_{rBC}$.

Third, the dependence of the ambient $MAC_{BC}$ on the coating composition is evaluated. Moffet et al. (2016) indicated that the chemical composition of the BC coating material may affect the optical properties of BC by influencing the location of a BC core within its hosting particle. These authors observed in a field study in central California that BC cores in particles with

organic-rich coatings were located near particle centers while BC cores in particles with inorganic coatings were located near particle edges. They hypothesized that the latter case occurs due to crystallization of the inorganic species. In terms of light absorption enhancement, a few laboratory and field studies have found evidence that $E_{MAC}$ values depend on whether BC is internally mixed with organic or inorganic species. Wei et al. (2013) found that glycerol-coated BC had $E_{MAC}$ at 532 nm of ~1.4 while BC cores coated with solid ammonium sulfate and ammonium nitrate had $E_{MAC}$ at 532 nm of only 1.10 and 1.06,

respectively. However, it should be noted that the amount of coating could not be quantified conclusively in these experiments, and therefore it cannot be verified that the coating volume fraction was equal for the different coating compositions. Zhang et al. (2018) applied a multi-linear regression analysis to positive matrix factorization (PMF) source apportionment results to conclude that highly oxidized secondary organic aerosol was the major chemical component responsible for aerosol light absorption enhancement observed at an urban background site in Paris, France. These authors showed that $E_{MAC}$ at 870 nm





displayed a positive relationship with the mass ratio of bulk secondary organic to secondary inorganic aerosol, with $E_{MAC}$ at 870 nm increasing from 1 to 2 as the mass ratio of organics to inorganic increased from 2 to 8. However, it should also be noted here that coating amounts were not measured, and therefore the observed relationship could potentially be the result of cross-correlation between coating composition and coating thickness.

In this study, the chemical composition was measured for bulk aerosols by an ACSM. No chemical or mixing state information is available at the single particle level. Therefore, we used the bulk ratio of measured inorganic to organic particulate matter as a proxy variable to investigate a possible dependence of $MAC_{BC}$ on coating composition. In Fig. 7c is seen that $MAC_{BC}$ displays no clear relationship with the inorganic to organic ratio. By contrast, darker points are systematically higher up than brighter points, indicating that the amount of coating (coating thickness) had a large effect on absorption enhancement through

the lensing effect, whereas variations in coating composition only had a negligible effect during this campaign. However, it cannot be ruled out that a relationship between coating chemical composition and absorption enhancement did in fact exist, but that the relationship is not apparent in Fig. 7c because the bulk ratio of inorganic to organic aerosol mass is a poor indicator of the composition of coatings on individual particles.





**3.5 Absorption enhancement factors (lensing effect) and their comparison with previous studies**

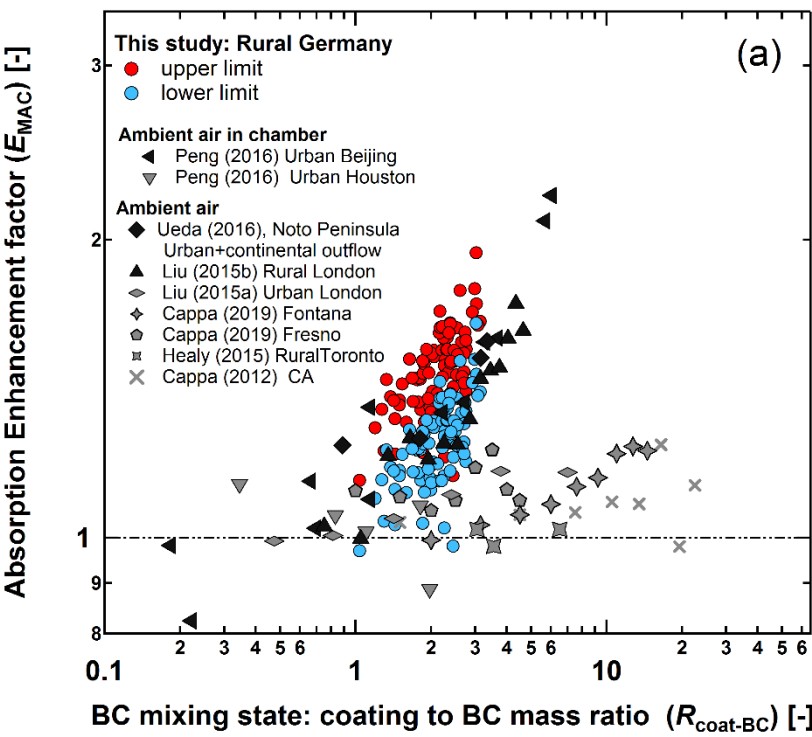


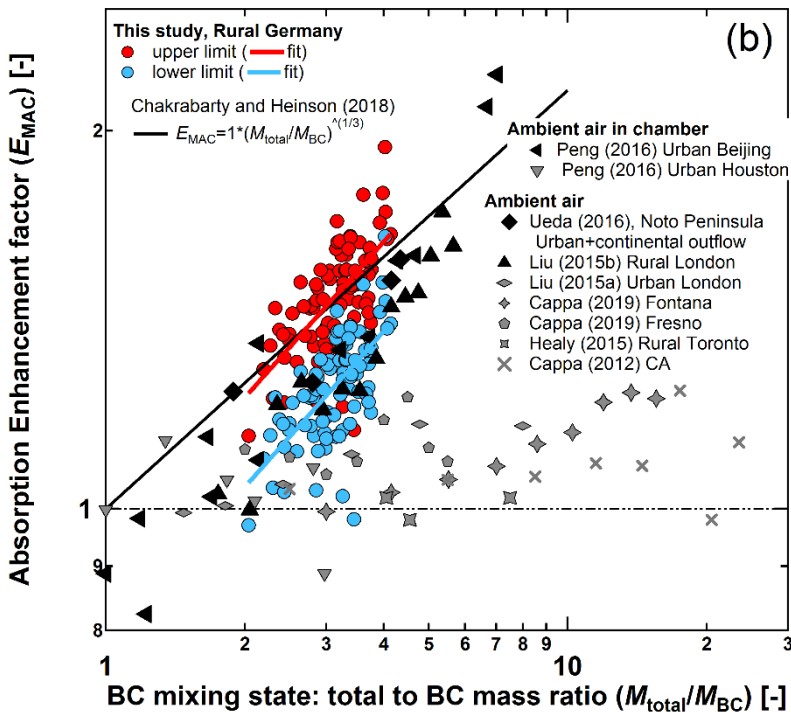

**Figure 8.** Summary of $E_{MAC}$ values and their dependence on BC mixing state from this study compared with literature data measured at wavelengths from 532 nm to 870 nm. The $E_{MAC}$ values are plotted versus the ratio of coating to BC core mass (a) and versus the ratio of total particle to BC core mass (b). Panel (a) is an updated version of a figure from Cappa et al. (2019) and panel (b) additionally includes the simple power-law dependence of the lensing effect proposed by Chakrabarty and Heinson (2018), which uses $M_{total}/M_{BC}$ as free parameter. The two approaches applied to obtain the lower and upper limit $E_{MAC}$ values presented for this study are discussed in Sect. 3.3). Note: the data points of the studies by Ueda et al. (2016) and Liu et al. (2015a) deviate marginally from those in the original figure by Cappa et al. (2019), as the abscissa values were recalculated from the original data using equal material densities for BC and coating material of 1.8 and 1.6 g cm$^{-3}$, respectively, as applied in this study.

The relationship between $E_{MAC}$ and the internal mixing state of BC is shown in Fig. 8a. In order to directly compare the results of this study with those from previous studies, the BC mixing state is represented in Fig. 8a by the mean ratio of coating to core mass ($R_{coat-BC}$) for BC cores with mass equivalent diameter between 200 and 220 nm, rather than mean coating thickness over this same size range as in Fig. 6. Two sets of data points from this study are shown in Fig. 8a: lower limit estimates of $E_{MAC}$ that were obtained by the denuding method, and upper limit estimates of $E_{MAC}$ that were obtained by the correlation method, as described in Sect. 3.3. The differences between these two sets of measurements is at least partly due to the fact that the thermodenuding process at 350 °C employed here was not sufficient to completely remove all coating material from the BC particles (Figs. 5c and S8). This is an important point to consider in all studies that employ similar types of thermodenuders in order to remove coatings from BC particles.



$E_{MAC}$ measurements from previous studies are also displayed in Fig. 8a in order to place the results from this study in context. The most striking aspect of this comparison is that ambient $E_{MAC}$ measurements tend to cluster into two main branches of points, as was recently pointed out by Cappa et al. (2019). One branch of measurements, indicated by light grey markers in Fig. 8a, suggest that $E_{MAC}$ has little or no relationship to $R_{coat-BC}$ (i.e. a weak lensing effect). By contrast, the second branch of measurements (black markers in Fig 8a) indicate a strong positive relationship between $E_{MAC}$ and $R_{coat-BC}$ (i.e. a clear lensing effect). The results from this study populate the latter branch of points, indicating the occurrence of a clear lensing effect.

The reasons why a strong lensing effect is observed in some ambient studies but not in others remain elusive and hypothetical. Firstly, it's important to note that there are methodological differences between the studies summarized in Fig. 8a. Some studies employed an SP-AMS to measure BC mixing state (Cappa et al., 2012; Cappa et al., 2019; Healy et al., 2015; Liu et al., 2015b), while other used alternative instruments such as SP2, transmission electron microscopy (TEM) and a V-TDMA (volatility tandem differential mobility analyzer) for this purpose (Liu et al., 2015a; Ueda et al., 2016; Peng et al., 2016 and this study). It is noteworthy that the majority of SP-AMS-based studies observed only a weak lensing effect, which suggests that this result may be related to instrument characteristics, such as the fact that the instrument collection efficiency depends on BC mixing state (as discussed in the Introduction section). However, a weak lensing effect was also observed in one SP2 based study (Liu et al., 2015a) and a strong lensing effect in one SP-AMS based study (Liu et al., 2015b), which suggests that the discrepancies between studies are not only due to this methodological reason. Nevertheless, dedicated SP-AMS and SP2 instrument inter-comparison experiments should be conducted to directly compare $R_{coat-BC}$ measurements from these two instruments to see to what extent discrepancies between the methods can explain the divergence of results in Fig. 8a.

Secondly, there are plausible physical explanations to explain the clustering of measurements into the two branches of points displayed in Fig. 8a, as discussed by Cappa et al. (2019). One hypothesis is that the distribution of BC and other aerosol components on a per-particle level determines whether a lensing effect occurs or not. Theoretical studies (Adachi et al., 2010; Zhang et al., 2017) and laboratory measurements (Schnaiter, 2005) indicate that BC particles encapsulated in a core-shell configuration display substantial absorption enhancement, while BC particles that are only partially encapsulated or 'attached' to the edge of other particles display little or no absorption enhancement. TEM studies have found that both of these types of mixed BC particles can be found in the ambient atmosphere (Ueda et al., 2016; Liu et al., 2015b), and some studies have even linked the presence of concentrically coated BC particles to higher observed $E_{MAC}$ (Ueda et al., 2016). Recent studies indicate that a threshold value of $R_{coat-BC}$ appears to exist beyond which particles collapse to a core-shell-like configuration and display a substantial lensing effect (Peng et al., 2016; Liu et al., 2017). Aside from individual-particle-level morphology, the distribution of coating material across an ensemble of particles is also an important determinant of the average absorption enhancement displayed by that ensemble (Fierce et al., 2016). This is another potential hypothesis for why some studies have measured lower than expected $E_{MAC}$ values for aerosols with large $R_{coat-BC}$.





During the Melpitz campaign, neither measurements of BC particle morphologies nor single-particle-level coating thickness measurements over the full size range of BC particles are available (representative LEO-fit measurements only cover a limited range, see Sect. 2.4.1). Therefore, we are unable to say definitively why a clear lensing effect was observed in this study and not in other studies. However, given the fact that a clear lensing effect was observed, we assume that the majority of BC particles were above the threshold of absorption enhancement and were fully coated in a core-shell like configuration. This in turn suggests that BC particles measured at the rural background site of Melpitz during winter were in a relatively aged state.

Recently, Chakrabarty and Heinson (2018) proposed that the absorption enhancement of BC particles as a function of the total particle to BC core mass ratio follows a simple power law with an exponent of 1/3. This finding is based on numerically exact electromagnetic calculations of simulated BC aggregates with variable degree of internal mixing with non-absorbing matter, and it is also in agreement with some previous experimental studies. The $E_{\mathrm{MAC}}$ results from this study and selected ambient literature studies are plotted against BC mixing state expressed as $M_{\mathrm{total}}/M_{\mathrm{BC}}$ in Fig. 8b. Our results and some of the other studies having clear lensing effect (black markers) are generally consistent with the proposed power-function scaling law. However, other studies having weak lensing effect (light gray markers) disagree with the scaling law, which could be due to the hypotheses mentioned previously. It is noteworthy that the upper limit $E_{\mathrm{MAC}}$ estimates of this study appear to more closely follow the scaling law than the lower-limit estimates. This may suggest that the upper limit $E_{\mathrm{MAC}}$ estimates are more realistic, which could be reasonable considering that the lower limit estimates were derived from the denuding measurements which failed to completely remove the BC coatings. However, quantitative comparison of ambient results should be treated with care, given the limited range of mass ratios that were observed and the potential influence of other minor factors on $E_{\mathrm{MAC}}$ as described in Sect. 3.4.

## 4 Conclusions

Field measurements of BC particle properties and additional aerosol characteristics were performed at the rural background site of Melpitz, Germany during winter (February 2017). Two independent methods (the denuding and correlation methods) were used to show that the variability of $\mathrm{MAC_{BC}}$ at this site was driven by the degree of BC internal mixing state (i.e. a clear lensing effect was observed). The enhancement of light absorption by BC due to coatings varied from 1.0 to 1.6 (lower limit estimates), or 1.2 to 1.9 (higher limit estimates), mean BC volume fractions in the internally mixed particles varying from 22 to 46 % (for BC core sizes from 200 to 220nm). The lower and higher limit estimates were determined using two different approaches, one of them involving denuding by means of a catalytic stripper, and the difference between the two may potentially be caused by incomplete coating removal.

The strong lensing effect observed in this study agrees well with a theoretical relationship recently published and with a subset of previous ambient studies. No evidence was found for cases with absence of lensing effect despite internally mixed BC, as





reported in other previous studies. Additional potential drivers of $MAC_{BC}$ variations including dominant BC source, average

690  BC core size and coating composition were also investigated. None of these was found to have a substantial effect. However, such effects could potentially be obscured by the lack of single particle information required for more quantitative assessment. Overall, the results of this study support that knowing the BC mixing state in terms of BC volume fraction in the internally mixed particles is sufficient to describe the lensing effect and $MAC_{BC}$ in good approximation. By contrast, the influence of coating composition appears to be minor for atmospheric aerosols. However, future field studies addressing the coating

695  composition effect would have to combine quantitative mixing state measurements with coating composition measurements, and also mixing state morphology, which remains an experimental challenge.

*Data availability.* Data will be made available on Zenodo if the manuscript was accepted for publication.

*Author contributions.* MGB acquired the funding, and he designed the experiment jointly with RLM and JY, and BW

700  coordinated the campaign. JY, RM, MZ, TM, LP, BW and TT took the measurements and/or analysed the raw data. JY interpreted the results and wrote the manuscript together with RLM and MGB. All co-authors reviewed and commented the manuscript.

*Competing interests.* The authors declare that they have no conflict of interest.

*Acknowledgements.* The authors gratefully acknowledge Achim Grüner and Gerald Spindler for regular operation of the

705  Melpitz observatory and the NOAA Air Resources Laboratory (ARL) for the provision of the HYSPLIT transport and dispersion model. Financial support was received from the ERC (grant agreement no. 615922-BLACARAT). Further support was received from the ACTRIS2 project funded by the EU (H2020 grant agreement no. 654109) and the Swiss State Secretariat for Education, Research and Innovation (SERI; contract number 15.0159-1). The opinions expressed and arguments employed herein do not necessarily reflect the official views of the Swiss Government. MZ and ABH gratefully acknowledge the funding

710  by the Deutsche Forschungsgemeinschaft (DFG, German Research Foundation) – Projektnummer 268020496 – TRR 172, within the Transregional Collaborative Research Center "ArctiC Amplification: Climate Relevant Atmospheric and SurfaCe Processes, and Feedback Mechanisms (AC)³".



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
