# Peer review of "Variability in the mass absorption cross-section of black carbon (BC) aerosols is driven by BC internal mixing state at a central European background site (Melpitz, Germany) in winter"

_Atmospheric Chemistry and Physics, 2020_

## Referee Comment (RC1) · Anonymous Referee #1 · 3 Mar 2020

General comments:

This study presents the mass absorption cross-section of BC and the related parameters based on observation at background site. They showed that clear correlation between coating thickness and mass absorption cross-section of BC. One of the relations between lensing effect and mass ratio of BC was well corresponding with simply relation based on recent simulation study which considered some morphological factors. These main results based on observation measurements are of interest and useful to the community. I found that the measurements were conducted and analyzed

well carefully. However, some evaluations for factors which concluded as "minor role" seemed to be biased and insufficiently. I felt that measurements and discussions was partly different from that expected in abstract and introduction. I recommend publication of their study after the following expressions are improved.

Specific comments:

1) "Mixing morphology" in abstract and introduction: "Morphology" sets reader expectation for shape factor, such as aggregation or chain, core-shell or non-spherical, attached or coating, and etc. However, this study did not investigate morphology itself (e.g. microscopic observation) and parameters well-relevant with shape (e.g. particle density and light polarization). In discussion, I found that authors used ratio of inorganic to organic in place of morphology, based on previous knowledge. However, as authors mentioned in discussion, ACSM data contains none of single particle information relating external and internal mixed. Also, organic species in a particle and the phase (liquid or solid) of the particle in the atmosphere relates to formation of the morphology. Therefore, it would be impossible in their measurements to infer and evaluate morphology. I recommend to replace the word into direct expression of measurements such as "ratio of inorganic to organic (in bulk particles)" simply. For whole manuscript, some "morphology" relating with mixing states should be revised. "mixing morphology" were used in introduction relating a hypothesis by Cappa et al. (2019), but I could not find the word the literature. The word is not general and gives confusion.

2) Introduction explained that maximal MAC depends on particle morphology and size. Although increase of coating thickness of BC can enhance the lensing effect, the lensing effect of atmospheric aerosols would be less than that expected by spherical core-shell shaped particles. In addition, coating thickness and morphology (morphology of individual particles and distribution of different morphological variation of particles) can alter in combination by aging process. These are not independent parameter. Inhibition of lensing effect by morphology and size can affect to not only correlation but also slope of relation between MAC and coating thickness. Therefore, the minor or major

roles should not be defined by only correlation. The observation period was short. It is unclear that morphological factors in the period changed such to have given variation of the EMAC.

3) All parameters for aerosols seemed to be measured after passing drier in the study. This point should be noted in abstract, discussion and conclusion. For example, hygroscopicity of BC-containing particle depends on coating composition. If the coating thickness strongly affect the lensing effect, the deliquescence relative humidity and growth factor depending on the composition can influence on the lensing effect.

4) I could not understand what specific cause and process was expected to different BC source as factor affecting to MACBC. Coating thickness usually increases with aging process. If MACBC is different according to BC source, the difference will be clearer freshly BC before aging. The coating thickness as shown in Fig. S7 was not bi-modal distribution which often found in urban sites, suggesting the BC observed in the site was almost well-aged. It is not surprising that effect by property of BC core in source decrease with increase coating thickness. I think that discussion of BC source should be mentioned with property of the observation site and the aging level of BC.

5) How was relation of MACBC after denuded BC coating to AAE and diameter of rBC? As commented above, I thought that it would be difficult to evaluate these relations after aging proceed.

Technical comment and minor issues:

Abstract:ãĂĂ As commented in specific comments, abstract should be improved for reader to understand contents which were directly used in measurement and evaluation. The conclusion should be limited adequately for atmospheric condition, considering the method and the property of observation site.

P2L54 "The maximal MAC enhancement factor that can be reached for a particle depends on particle morphology and size, with greater values for smaller particles." Which

are these "particle" meaning "BC core/BC particle" or "BC-containing particle"? The sentence sound not right if they are BC-containing particle because particle morphology and size can alter by aging process.

Figure S1: What does mean the "all particles"? I wondered about their inconsistence with sum of BC-free particle and BC-containing particle.

P9L1 "choosing it in this manner ensures bias-free measurements of the coating thickness of uncoated bare BC particles." I could not understand this sentence until section 2.4.3. (I confused why "uncoated bare BC" have coating?). Also, in this paragraph, "bare BC", "BC core", and "uncoated BC" was used, but their difference was unclear.

P11L309 "Since the MAAP is more robust as an absolute reference,. . ." I cannot agree with this sentence. MAAP measure absorption of cumulative particles on filter. Therefore, the absorption might include more inaccurate lensing effect comparing to that by photo-acoustic spectroscopy.

Table S1, ~P12L330 Please specify instrument information of SO2, NOx and aerosol concentration, and species measured by ACSM, in Table S1 or section 2.2.

P15L394 Probably, "Fig. 05b" is "Fig. S5b".

P20LL498 "It is seen that the denuded-MACBC values all fall in the range from 5.6 to 6.0 m2 g-1. . ." However, some yellow dots seem to be >7 m2 g-1, which were probably in gray period of Figure 5. Does the MACBC of denuded BC depend on coating of denuded BC coating?

P20LL461 Although authors inferred short residence time as a cause of incomplete remove coating. However, some residuals such as incomplete charring of organic compounds can also remain after passing denuder at 350âŮęC. The absorption of such charring organics would be slight at 870 nm. However, I think that lensing effect by residuals can appear as some bias of MACBC of denuded BC coating when such residuals presence on BC core.

P26L616 Sorry If I miss the point. Is upper limit of EMAC are MACBC,ambient/5.8? Which is upper limit of EMAC, MACBC,ambient/5.0 (constant) or MACBC,ambient/ MACBC,denuded (time variable)? I recommend to show equations.

P28L668 Which instruments measured the Mtotal and MBC? What particle dose "total" contain? BC-containing particles or whole partilces?

Figure 8 Please remove "BC mixing state" of x-label because of above confusion.

---

## Referee Comment (RC2) · Anonymous Referee #2 · 2 Apr 2020

The authors report on measurements of the relationship between absorption by black carbon and the extent to which the BC particles are coated. Overall, I find this is a nice addition to the existing literature on this topic, and the results appear to be of high quality. I have a number of comments that I think the authors should address before this is accepted for publication. I would ask that they pay particular attention to the comments regarding Fig. 8 as there are a number of aspects that I find unclear about the data presented in this figure.

L40: The authors might more accurately state that when BC is freshly emitted it "may

be" separated from other species. Some combustion processes emit BC already internally mixed with some other components (e.g. organics). Also, in the next sentence I suggest it would be more precise to say that "particles" coagulate (not "species").

L47: I'm not certain that Mie theory deals with "refraction" of light, as the authors state. "Refraction" is more of a geometric optics concept.

L70: The authors might note that the results of Qiu et al. (2012) are outliers among the now numerous lab experiments that indicate notable enhancements occur for size-selected BC.

L80: It seems a bit of a stretch to me to simply state that the SP-AMS is not quantitative owing to variability in detection efficiency, with citing only of the Taylor et al. (2015) paper. This effect has been noted by others in papers that focused on this issue (Willis et al., 2014) and others have used this knowledge to account for the variability in the detection efficiency (e.g., Collier et al., 2018). (I'll also note that Taylor et al. show the SP-AMS/SP2 BC ratio versus the absolute concentration of inorganic species measured by the SP-AMS, not versus the coating-to-BC ratio as implied by the authors use of the term "mixing state" here.) I suggest the authors temper the statement here a bit to indicate that quantification is challenging and care must be taken to account for changes in detection efficiency that might occur as the coating state of the BC changes. Also, it would be useful if the authors would clarify whether they are using "absolute" here to mean the absolute concentrations or the absolute coating thickness. I believe they mean the latter, based on the discussion in the next paragraph. However, this seems to contradict, somewhat, the authors statement that the SP-AMS does well with the coating-to-core ratio, which is the primary determinant of the coating thickness (if one translates from a mass ratio to a coating amount).

L87: Given the citation of the Taylor et al. (2015) paper above, it seems appropriate that the authors here might acknowledge some of the challenges in extracting absolute measures of the coating thickness that result from having to make particular assumptions regarding the BC material properties. (This does come up later, briefly.) Additionally, the SP2 interpretation method inherently assumes spheres yet the particles may not be spherical (as the authors note above), which will affect the accuracy of the method; it seems this should be noted. Also, the authors cite here Laborde et al. (2012a) as justification for "quantitative" coating thickness determination. It is unclear where in that paper coating thickness determination is shown to be quantitative; the paper seems to actually be quite qualitative in terms of coatings. As such, I suggest this reference be clarified, removed, or replaced with a more appropriate reference.

L115: Is it quite correct to state that the MAC is compared to the "BC particle mixing state?" Coating thickness is not exactly mixing state. At minimum, the authors should clarify that they mean "internal mixing state" per their discussion earlier. But I suggest that rather than using "mixing state" here and "infer[ring]" mixing state from coating thickness they just state that they compare with mean coating thickness for particles in a particular size range from the SP2. I feel that the use of the term "mixing state" here serves to confuse rather than clarify, at least for me. The same I find true when the authors state that they determine "mixing state" by thermodenuding. I suggest just stating what specifically was measured, for example (L120) that the authors aim to infer a causal relationship between the lensing effect and the "coating amount" rather than "mixing state."

L192: The authors note that the SP2 "missing mass correction" will be detailed in a separate paper that is not available for review at this time. Therefore, I suggest that the authors provide at least a short summary of the correction method. For example, was a single campaign average value applied, or did the authors determine a missing mass correction based on three hour averages (as this is their averaging time for the various measurements)? If the latter, noting min/max values (as percents) would seem appropriate. It seems to me that the latter is more appropriate. Certainly, based on the mean values given the missing mass correction details should have small effect, but they should nonetheless be noted here.

SP2 coating determination: While the details provided here are most certainly important, the authors could probably move most of these details to the supplemental and then note more succinctly in the main text the method used and the key uncertainties.

L211: It is unclear to me how Fig. S1 shows that optical diameters are within "a few percent" of the mobility diameters. Are the authors making this statement based on some general similarity in shape? Also, in Fig. S1 I find it unclear whether the red curve is the sum of the blue and the black, as it should be based on the definitions provided. If it is the sum, this means that the entirety of the blue curve is not shown. The authors might consider visual ways to clarify.

The minimum detectable coating thickness seems to come from Fig. S3b. I suggest that the authors report the +/- 10% and the 10th and 90th percentile bounds as diameter equivalent. I think the +/- 10% line is nominally 10 nm and the 10th/90th percentile is 30 nm.

L305: A minor issue, but the authors might clarify whether the AAE values here are the concurrently measured 3-h averages.

L308: It would be helpful if the authors clarify how a 31% bias translates to a scaling factor of 1.44. Why not 1.31?

Fig. 2: Visually, it appears that the difference between the 1:1 line and the scaled measurements is largest at intermediate absorption values. It would be useful if the authors were to consider the percent difference as a function of absolute absorption, and comment on how this might (or might not) impact their conclusions here.

L322: Since the losses are size dependent, and since coating-to-BC ratios are likely size dependent as is the SP2 detection efficiency, the authors might instead state that the losses likely introduced minimal bias, rather than stating categorically that they introduced no bias.

L323: The shapes of the particles may have changed upon denuding. Might this impact

the determination of coating thickness, as the method assumes spherical particles (even for the BC)?

L416: It is not clear to me how Fig. 2 indicates the lower limit of quantification for the PAX. It just shows the relationship between the PAX and MAAP. The limit of quantification typically comes from consideration of the instrument noise when sampling zero particles over the time period of the instrument zeros.

L438: It would be helpful if the authors could clarify why they are not considering emissions from biomass combustion, which tend to also produce larger BC cores, instead focusing on coal as the major BC source.

L468: Do the results here change if the authors instead use the median values (rather than the mean)? Or, what if the authors use a geometric average rather than an arithmetic average. The former is typically more appropriate for distributions that are bounded on one side. Here, the coating thickness cannot be <0 nm (within error), and thus the distributions are inherently non-Gaussian but instead more log normal. (Also, Fig. S7 reports the medians. It would be good to see things reported in a common way.)

Fig. 6: Given that the authors use the standard error of the mean to show their uncertainties here, it would be helpful if they would report the typical instrumental averaging times so that the reader can know how many points go into the 3-h averages. (In other words, are these averages of 1s data? 1 min? 10 min?)

Fig. 6b: Given that the denuded particles appear to still have coatings, I suggest that it would be helpful to show also a composite plot with the ambient particles versus their coating thickness and the denuded particles versus their coating thickness so that the continuity between these can be seen.

L497: It is not accurate to state that "all" the denuded MAC values fall in the stated range. There are some points that are outside this range. "Most" of the denuded MAC

values fall in the stated range.

L549: Biomass burning emissions also exhibit a wide range of AAE values. Thus, it is not clear why apportionment is appropriate in the biomass burning assumption above but not for coal. It is also not clear why the biomass burning method would provide an "upper limit estimate" for the contributions. How is it specifically known that this is an upper limit?

L 562: I suggest it would be useful for the authors to calculate and report the R2 value when the two clear outlier points are excluded. These correspond to the short period when the BC source was quite different. The R^2 in this case would definitely be >0.01, although still not as large as the relationship with the coating thickness shown previously. Also, I find it a little awkward to note that variability in the BC diameter over the range 150 nm to 250 nm is a "narrow range" but the coating thicknesses, that only varied from ~40-60 nm is not "narrow." Also, visually there appears some correlation between the BC diameter and the coating thickness from Fig. 5 (excluding the short plume). Finally, from the size distributions shown in Fig. S5, it would seem that any missing mass correction would be larger for period 2 than for period 3 than for period 1. To what extent might uncertainty in the missing mass correction contribute to the relationships shown? The distributions are close to log-normal, but not quite and thus use of a single mode fit might underestimate the correction that is necessary.

L567: The authors might note that this is the theoretical result of Mie theory, which might not be fully applicable to fractal-like particles for which absorption by the individual spherules might dominate.

L578: I would suggest the authors go a bit further and argue that it is very likely that the coating amounts differed, as no attempt was made in that study to ensure that the particles had the same amount of coating.

Fig. 8 and origin of Ueda et al. (2016) data: The authors show results from Ueda et al. (2016) in this figure. In their caption they note that the points shown might differ

from those shown in Cappa et al. (2019). However, in looking at Cappa et al. (2019) the Ueda et al. data are not included. As such, it is unclear where these data come from. Looking at the Ueda et al. (2016) paper directly, there is no indication that values of R_coat-BC are available in a general sense. At best, it would seem that up to four data points might be included, corresponding to the periods A-D in Ueda et al. (2016). It looks like the authors used the mean volume fraction of soot for particles in varying size ranges and with varying number of particles analyzed (per Ueda et al., Table 3) to calculate the volume fractions. For one of these periods (C), only 6 particles were analyzed, calling into question the statistical significance of any volume fraction. As for the Eabs values, did the authors use the 400 degC results reported? Or did they calculate an estimated Eabs based on the reported [BC] and absorption at 781 nm? I suspect the former, but it might be noted that if one assumes a constant MAC for uncoated BC and calculates an Eabs from the reported measurements, the thus derived Eabs does not match with the value derived from heating to 400 degC in terms of their apparent dependence on the coating volume fraction.

Fig. 8b: It is unclear whether the x-axis is really the total NRPM-to-BC ratio for all studies shown. For some of the studies, the x-axis values in panel b appear the same as in panel a, which cannot be the case unless all NRPM is internally mixed with BC. But for other studies the relationship differs. Additionally, it is also not clear for the current measurements that the x-axis is correct. Looking at Fig. 2, the BC fraction ranges from about 0.04 to 0.22, excluding the short plume. The NRPM/BC ratio should just be (1-BCfraction)/BCfraction, and so should range from approximately 3.5 to 24. But the data shown in Fig. 8b only range from 2-4 for the current study. I suggest that clarification is needed. Perhaps I am just misunderstanding the relationship between the R_coat-BC and M_total/M_BC as used here. Or misunderstanding what the authors mean in the figure caption when they note that the M_total/M_BC is a "free parameter." Free in what way? (As a minor note, inclusion of ticks between 1 and 2 on the y-axis would be helpful to the reader.)

[Figure]

L638: I suggest it would be helpful for the authors to be more explicit about the SP-AMS here. The introduction section does not indicate that the coating-to-core ratio from the SP-AMS should depend strongly on the coating amount but instead notes that absolute quantification of concentrations depends on the coating amount. How does a change in the collection efficiency affect the ratio, rather than the absolute values? Also, they might note that the laboratory studies that have looked at this effect find that above a coating-to-core ratio of ~3 that the collection efficiency is effectively constant, and some of the studies shown do have coating-to-core ratios this large.

L640: This is a really great point, that I suggest the authors re-emphasize specifically in their conclusions.

In their discussion of physical explanations for the differences between studies, the authors might note that some of the studies are more likely to be impacted by local sources and some by particles that have undergone long-range transport. The latter might tend to homogenize the population, which could affect the observable absorption enhancement.

L681: The sentence beginning "The enhancement..." does not seem to be a complete sentence.

L691: It is not overly clear to me how "such effects" might be obscured here. What do the authors mean when they indicate "a more quantitative assessment?" Do they mean calculated across the entire size distribution?

References

Collier, S., Williams, L. R., Onasch, T. B., Cappa, C. D., Zhang, X., Russell, L. M., Chen, C.-L., Sanchez, K. J., Worsnop, D. R., and Zhang, Q.: Influence of emissions and aqueous processing on particles containing black carbon in a polluted urban environment: Insights from a soot particle – aerosol mass spectrometer, Journal of Geophysical Research-Atmospheres, 123, 6648-6666, https://doi.org/10.1002/2017JD027851,

2018. Taylor, J. W., Allan, J. D., Liu, D., Flynn, M., Weber, R., Zhang, X., Lefer, B. L., Grossberg, N., Flynn, J., and Coe, H.: Assessment of the sensitivity of core / shell parameters derived using the single-particle soot photometer to density and refractive index, Atmos. Meas. Tech., 8, 1701-1718, https://doi.org/10.5194/amt-8-1701-2015, 2015. Willis, M. D., Lee, A. K. Y., Onasch, T. B., Fortner, E. C., Williams, L. R., Lambe, A. T., Worsnop, D. R., and Abbatt, J. P. D.: Collection efficiency of the soot-particle aerosol mass spectrometer (SP-AMS) for internally mixed particulate black carbon, Atmospheric Measurement Techniques, 7, 4507-4516, https://doi.org/10.5194/amt-7-4507-2014, 2014.

---

## Author Comment (AC1) · 1 Sep 2020

We thank the editor and reviewers for the timely handling of our manuscript, particularly during these difficult corona times. Please find below our point-by-point responses to each of the reviewer comments, including descriptions of the modifications we have made to the manuscript. Reviewer comments are in black text and our responses are given in blue text. Line numbers refer to the modified version of the manuscript with track changes highlighted.

**Anonymous Referee #1**

**General comments:**

This study presents the mass absorption cross-section of BC and the related parameters based on observation at background site. They showed that clear correlation between coating thickness and mass absorption cross-section of BC. One of the relations between lensing effect and mass ratio of BC was well corresponding with simply relation based on recent simulation study which considered some morphological factors. These main results based on observation measurements are of interest and useful to the community. I found that the measurements were conducted and analyzed well carefully. However, some evaluations for factors which concluded as "minor role" seemed to be biased and insufficiently. I felt that measurements and discussions was partly different from that expected in abstract and introduction. I recommend publication of their study after the following expressions are improved.

We thank the referee for his careful review and constructive comments, which we addressed as explained in the responses given below.

**Specific comments:**

1) "Mixing morphology" in abstract and introduction: "Morphology" sets reader expectation for shape factor, such as aggregation or chain, core-shell or non-spherical, attached or coating, and etc. However, this study did not investigate morphology itself (e.g. microscopic observation) and parameters well-relevant with shape (e.g. particle density and light polarization). In discussion, I found that authors used ratio of inorganic to organic in place of morphology, based on previous knowledge. However, as authors mentioned in discussion, ACSM data contains none of single particle information relating external and internal mixed. Also, organic species in a particle and the phase (liquid or solid) of the particle in the atmosphere relates to formation of the morphology. Therefore, it would be impossible in their measurements to infer and evaluate morphology. I recommend to replace the word into direct expression of measurements such as "ratio of inorganic to organic (in balk particles)" simply. For whole manuscript, some "morphology" relating with mixing states should be revised. "mixing morphology" were used in introduction relating a hypothesis by Cappa et al. (2019), but I could not find the word the literature. The word is not general and gives confusion.

We agree with the referee that "morphology" was used in an imprecise manner in some places. All instances of "morphology" where checked and where needed replaced with more specific terms.

2) Introduction explained that maximal MAC depends on particle morphology and size. Although increase of coating thickness of BC can enhance the lensing effect, the lensing effect of atmospheric

aerosols would be less than that expected by spherical coreshell shaped particles. In addition, coating thickness and morphology (morphology of individual particles and distribution of different morphological variation of particles) can alter in combination by aging process. These are not independent parameter. Inhibition of lensing effect by morphology and size can affect to not only correlation but also slope of relation between MAC and coating thickness. Therefore, the minor or major roles should not be defined by only correlation. The observation period was short. It is unclear that morphological factors in the period changed such to have given variation of the EMAC.

Figures 4 and 7 show that different air mass transport patterns and a considerable range in BC particle properties and sources were observed, suggesting that a fair amount of variability expected in winter time at this rural background site was likely covered during the observation period. We added the qualifier "…rural background site…" in the abstract:
"These results for ambient BC measured at Melpitz during winter show that the lensing effect caused by coatings on BC is the main driver of the variations in $MAC_{BC}$ and $E_{MAC}$, while changes in other BC particle properties such as source, BC core size or coating composition play only minor roles at this rural background site."
and in the conclusions
"…No evidence was found for cases with absence of lensing effect despite internally mixed BC, as reported in other previous studies. Additional potential drivers of $MAC_{BC}$ variations including dominant BC source, average BC core size and coating composition were also investigated. None of these was found to have a substantial effect at the rural background site Melpitz. However, such effects could potentially be obscured by the lack of single particle information required for more quantitative assessment….".

To what extent can morphology and size effects on the MAC potentially be hidden in a reduction of the slope of the correlation between MAC and coating thickness? Figure 7b suggests that the MAC is largely independent of the modal diameter of the BC core mass size distribution for a constant coating thickness (points of similar color are approximately horizontally aligned). By contrast, MAC varies systematically with coating thickness (the darker the points the higher up). Therefore, we retain our statement that the effects of coating thickness variations on MAC variations clearly dominate the effects of BC core size variations. As for the morphology: lacking particle morphology measurements we cannot directly exclude that BC particle morphology effects strongly affect the slope of the correlation between MA C and coating thickness. However, there is indirect evidence, as explained in the revised manuscript:

In Section 3.5: "Nevertheless, the agreement of the observation of this study with a model based on coated BC particles with morphologies favorable for occurrence of the lensing effect provides indirect evidence that dampening of the lensing effect due to particle shapes with unfavorable morphologies was only minor or negligible for the rural background aerosol at Melpitz in winter."

In the abstract and conclusions: "Indirect evidence suggests that potential dampening of the lensing effect due to unfavorable morphology was most likely small or even negligible."

3) All parameters for aerosols seemed to be measured after passing drier in the study. This point should be noted in abstract, discussion and conclusion. For example, hygroscopicity of BC-containing particle depends on coating composition. If the coating thickness strongly affect the lensing effect, the

deliquescence relative humidity and growth factor depending on the composition can influence on the lensing effect.

Important point. We have made additions to abstract, conclusions, captions of Figures 5&6 and in Section 3.3 in order to emphasize that all measurements are done for ambient aerosol dried to low RH.

4) I could not understand what specific cause and process was expected to different BC source as factor affecting to MACBC. Coating thickness usually increases with aging process. If MACBC is different according to BC source, the difference will be clearer freshly BC before aging. The coating thickness as shown in Fig. S7 was not bi-modal distribution which often found in urban sites, suggesting the BC observed in the site was almost well-aged. It is not surprising that effect by property of BC core in source decrease with increase coating thickness. I think that discussion of BC source should be mentioned with property of the observation site and the aging level of BC.

"…Central European Background site…" is already mentioned in the title and we have added the qualifier "…at this rural background site with a large fraction of aged particles…" to abstract and conclusions in order to emphasize the aged nature of the aerosol.

We also added the following paragraph at the end of Sect. 3.4:
"Despite the fact that atmospheric aging processes tend to make aerosols more homogeneous during transport away from sources, e.g. by increasing the degree of internal mixing, some dependence of $MAC_{BC}$ on BC source could be retained. Possible drivers for such source dependence include differences in BC core size and morphology, in chemical microstructure of the BC, in morphology of the coated BC due to differences in coating processes and/or coating composition. The $MAC_{BC}$ of denuded aerosol samples exhibited very limited variability (Fig. S4b) suggesting that potential differences in $MAC_{BC}$ of bare cores from different sources are small. The analyses presented above further suggest that BC source related differences in $MAC_{BC}$, which may or may not exist close to sources, largely disappear during transport to the rural background site. The only exception are variations in coating thickness, which have been shown to drive $MAC_{BC}$ variations, and which may have retained some residual relation with BC source."

5) How was relation of MACBC after denuded BC coating to AAE and diameter of rBC? As commented above, I thought that it would be difficult to evaluate these relations after aging proceed.

See response to previous comment.

**Technical comment and minor issues:**

Abstract:

As commented in specific comments, abstract should be improved for reader to understand contents which were directly used in measurement and evaluation. The conclusion should be limited adequately for atmospheric condition, considering the method and the property of observation site.

We have modified the abstract and conclusions as suggested so that the statements are limited appropriately to the measurement site and methods.

P2L54 "The maximal MAC enhancement factor that can be reached for a particle depends on particle morphology and size, with greater values for smaller particles." Which are these "particle" meaning "BC core/BC particle" or "BC-containing particle"? The sentence sound not right if they are BC-containing particle because particle morphology and size can alter by aging process.

The statement refers to BC cores. We have modified the sentence on L65 to clarify this: "The maximal MAC enhancement factor that can be reached for a BC core depends on particle morphology and size, with greater values for smaller cores"

Figure S1: What does mean the "all particles"? I wondered about their inconsistence with sum of BC-free particle and BC-containing particle.

'All particles' indeed refers to the sum of the BC-free and BC-containing particles. The confusion results from the fact that the BC-free distributions are not shown completely because they overlap with the BC-containing distributions. This was also commented on by Reviewer 2. We have replaced Fig. S1 with a version that makes this more clear by making the distributions semi-transparent so that the entire BC-free size (blue) size distributions can be seen. In addition, we modified the legend so that it is explicitly stated that the 'all particles' distributions are the sum of the corresponding 'BC-free' and 'BC-containing' distributions.

New version of Fig. S1:

[Figure]

P9L1 "choosing it in this manner ensures bias-free measurements of the coating thickness of uncoated bare BC particles." I could not understand this sentence until section 2.4.3. (I confused why "uncoated bare BC" have coating?). Also, in this paragraph, "bare BC", "BC core", and "uncoated BC" was used, but their difference was unclear.

We have modified the text in this paragraph to clarify and make consistent the use of the term 'bare BC core'.

Specifically, on L303 we have changed 'BC core' to 'bare BC core'.

And on L309 we have added a statement in parentheses to clarify the meaning of the sentence: "However, choosing it in this manner ensures bias-free measurements of the coating thickness of bare

BC particles (i.e., this choice results in coating thickness histograms for bare BC particles that are centered around 0 nm)."

P11L309 "Since the MAAP is more robust as an absolute reference,: : :" I cannot agree with this sentence. MAAP measure absorption of cumulative particles on filter. Therefore, the absorption might include more inaccurate lensing effect comparing to that by photo-acoustic spectroscopy.

Section 2.4.2.2 was entirely rewritten in order to clarify our approach to measure the absorption coefficient at high time resolution:
"The absorption coefficient was quantified with a combination of PAX, MAAP and AE33 data. The absorption coefficient measured by the MAAP at λ=637 nm was adjusted to λ=870 nm, using the AAE obtained from the AE33:

$$b_{\mathrm{ap,MAAP,870nm}} = b_{\mathrm{ap,MAAP,637nm}} \times (637/870)^{\mathrm{AAE}(637,870)} \tag{7}$$

The correlation between wavelength-adjusted MAAP data and PAX data is excellent in the range above ~1 Mm$^{-1}$ (Fig. 2). However, the $b_{\mathrm{ap,PAX,870nm}}$ measurements are biased systematically lower than the $b_{\mathrm{ap,MAAP,870nm}}$ measurements by 31%, which is most likely a result of imprecise and hence inaccurate PAX calibration (Section 2.4.2.1). The MAAP demonstrated good accuracy in several intercomparison studies and it has been shown to have a low instrument-to-instrument variability of less than 5 % (Müller et al., 2011). Therefore, we decided to scale the PAX data by a constant scaling factor of 1.44 [=1/(1-0.31)]. to match $b_{\mathrm{ap,MAAP,870nm}}$ as shown in Fig. 2. Application of this constant scaling factor brings the scaled PAX and MAAP measurements into good agreement at absorption coefficients greater than ~15 Mm$^{-1}$, while the scaled PAX measurements are up to 10% lower than the corresponding MAAP measurements in the range down to 1 Mm$^{-1}$ (as shown by the green crosses in Fig. 2). The scaled PAX data provide absorption coefficients with high time resolution and for both ambient and denuded inlets, with absolute calibration referenced to the MAAP."

Table S1, _P12L330 Please specify instrument information of SO2, NOx and aerosol concentration, and species measured by ACSM, in Table S1 or section 2.2.

We have modified the relevant parts of Sect. 2.2 to include this information.

L180: "The ACSM (Aerodyne Research, MA, US; Ng et al., 2011) measured the near-PM1 bulk chemical composition of non-refractory aerosol species including organics (Org), nitrate ($NO_3$), sulfate ($SO_4$), ammonium ($NH_4$) and chloride (Chl).  The ACSM measurements are described as near-PM1 since the instrument inlet has an upper cut-off at an aerodynamic diameter of around 1 µm."

L194: "Concentrations of $SO_2$ were measured with a UV-Fluorescence instrument (Type APSA 360A, HORIBA Jobin Yvon GmbH, Germany) and NO and $NO_2$ ($NO_x$) concentrations were measured with a Trace Level NOx Analyzer (Type 42i-TL, Thermo Fischer Scientific GmbH, Germany)."

P15L394 Probably, "Fig. 05b" is "Fig. S5b".

Yes, thanks for picking this up. Change made as suggested.

P20LL498 "It is seen that the denuded-MACBC values all fall in the range from 5.6 to 6.0 m2 g-1: : :"
However, some yellow dots seem to be >7 m2 g-1, which were probably in gray period of Figure 5. Does the MACBC of denuded BC depend on coating of denuded BC coating?

Firstly, we incorrectly stated that all values fall within the range from 5.6 to 6.0 m2 g-1 (as also noted by Reviewer 2). We have modified the statement to state "…that most of the denuded-$MAC_{BC}$ values fall…" in the stated range (L618).

Secondly, we added a new figure to the supplementary information (Fig. S4) to explicitly show the relationship between the denuded MAC_BC values and the coating thickness of the denuded particles. It is seen that there is no clear correlation between these two variables (Pearson r = -0.29).

The full statement beginning L618 now says: "It is seen that most of the denuded-$MAC_{BC}$ values fall in the range from 5.6 to 6.0 $m^2 g^{-1}$ (median= 5.8 $m^2 g^{-1}$) and that the values are largely independent of the original coating thickness, with a Pearson correlation coefficient of only 0.02 (the denuded-$MAC_{BC}$ values are also largely independent of the coating thickness of the denuded particles as shown in Fig. S4, with a Pearson correlation coefficient of -0.29)."

P20LL461 Although authors inferred short residence time as a cause of incomplete remove coating. However, some residuals such as incomplete charring of organic compounds can also remain after passing denuder at 350 degC. The absorption of such charring organics would be slight at 870 nm. However, I think that lensing effect by residuals can appear as some bias of MACBC of denuded BC coating when such residuals presence on BC core.

Actually there is no lensing effect apparent when plotting the denuded-MAC_BC values against the coating thickness of the denuded particles, as discussed in the comment above. We made the statement that the residual coatings on the denuded particles "might still be responsible for a lensing effect" not because we have direct evidence for it, but because we cannot rule it out.

We have modified the statement in question on L628 to try and clarify these points: "The denuded particles retain thin coatings which might still be responsible for a lensing effect. However, such a lensing effect is not apparent when plotting the denuded-$MAC_{BC}$ values against the coating thickness of the denuded particles (Fig. S4). In any case, we consider the median denuded $MAC_{BC}$ of 5.8 $m^2 g^{-1}$ to represent an upper limit estimate of $MAC_{BC,bare}$"

P26L616 Sorry If I miss the point. Is upper limit of EMAC are MACBC,ambient/5.8? Which is upper limit of EMAC, MACBC,ambient/5.0 (constant) or MACBC,ambient/ MACBC,denuded (time variable)? I recommend to show equations.

Thanks for pointing this out we agree it is useful to clarify explicitly how these quantities were calculated. We have modified the statement on L767 as follows: "Two sets of data points from this study are shown in Fig. 8a: lower limit estimates of $E_{MAC}$ that were obtained with constraining the $MAC_{BC}$ of

bare BC cores by the denuding method (i.e., $E_{MAC}$ = MAC$_{BC,amb}$/5.8 m$^2$ g$^{-1}$), and upper limit estimates of $E_{MAC}$ that were obtained with constraining the MAC$_{BC}$ of bare BC cores by the correlation method (i.e., $E_{MAC}$ = MAC$_{BC,amb}$/5.0 m$^2$ g$^{-1}$), as described in Sect. 3.3."

P28L668 Which instruments measured the Mtotal and MBC? What particle dose "total" contain? BC-containing particles or whole partilces?

Different instruments were applied in different studies. We only considered studies that quantified non-BC matter internally mixed with BC and BC mass. See next item for clarification of M$_{total}$.

Figure 8 Please remove "BC mixing state" of x-label because of above confusion.

M$_{total}$ refers to the total particle mass of BC-containing particles. Reviewer 2 also had comments about this quantity and other clarification issues with respect to Fig. 8b. In response to these comments we have made the following modifications to make clarify precisely what this quantity represents and how it related to Rcoat-BC plotted in Fig. 8a.

The caption of Fig. 8 now states: "Figure 8. Summary of $E_{MAC}$ values and their dependence on BC mixing state from this study compared with literature data measured at wavelengths from 532 nm to 870 nm. The $E_{MAC}$ values are plotted versus the ratio of coating to BC core mass (a) and versus the ratio of total particle mass to BC core mass (only considering BC-containing particles) (b). Panel (a) is an updated version of a figure from Cappa et al. (2019), while in panel (b) the abscissa has been changed to $M_{total}$/$M_{BC}$ to additionally include the simple power-law parameterization of the lensing effect proposed by Chakrabarty and Heinson (2018), which uses $M_{total}$/$M_{BC}$ as the input parameter to represent BC mixing state (where $M_{total}$ refers to the total mass of the BC-containing particle, such that $M_{total}$/$M_{BC}$ = 1 + $R_{coat-BC}$). The two approaches applied to obtain the lower and upper limit $E_{MAC}$ values presented for this study are discussed in Sect. 3.3). Note: the data points from the study of Liu et al. (2015a) deviate marginally from those in the original figure by Cappa et al. (2019), as the abscissa values were recalculated from the original data using material densities for BC and coating material of 1.8 and 1.6 g cm$^{-3}$, respectively, as applied in this study."

And on L830 of the main text: "The $E_{MAC}$ results from this study and selected ambient literature studies are plotted against BC mixing state expressed as $M_{total}$/$M_{BC}$ in Fig. 8b (where $M_{total}$/$M_{BC}$ = 1 + $R_{coat-BC}$)."

**Anonymous Referee #2**

**General comments:**

The authors report on measurements of the relationship between absorption by black carbon and the extent to which the BC particles are coated. Overall, I find this is a nice addition to the existing literature on this topic, and the results appear to be of high quality. I have a number of comments that I think the authors should address before this is accepted for publication. I would ask that they pay particular attention to the comments regarding Fig. 8 as there are a number of aspects that I find unclear about the data presented in this figure.

We thank the reviewer for his/her assessment of our manuscript and the comments provided. We have endeavoured to suitably address each of these comments. We believe this has helped greatly to improve the manuscript by clarifying some important aspects. We took particular care with Fig. 8 and hope the discussion around this figure is now clearer.

**Specific comments:**

L40: The authors might more accurately state that when BC is freshly emitted it "may be" separated from other species. Some combustion processes emit BC already internally mixed with some other components (e.g. organics). Also, in the next sentence I suggest it would be more precise to say that "particles" coagulate (not "species").

We agree that this is an important qualifier. We have added the word 'often' to indicate that this is not always the case. We also modified the sentence following this to further clarify that we are referring to particles coagulating.

L50: "During the atmospheric aging of BC, non-BC particles coagulate (e.g. particulate sulfate, nitrate, organics) with or gaseous species condense onto BC particles to form a variety of internal mixing states."

L47: I'm not certain that Mie theory deals with "refraction" of light, as the authors state. "Refraction" is more of a geometric optics concept.

We agree that the use of the term 'refraction' here is potentially confusing given the precise definition of the term in geometric optics. We have changed 'refracted' to 'focused' on L58.

L70: The authors might note that the results of Qiu et al. (2012) are outliers among the now numerous lab experiments that indicate notable enhancements occur for size selected BC.

Agreed. We have added the following sentence to the end of this paragraph (L86): "It should be noted that the results of Qiu et al., (2012) are outliers among the more numerous laboratory studies showing notable absorption enhancements."

L80: It seems a bit of a stretch to me to simply state that the SP-AMS is not quantitative owing to variability in detection efficiency, with citing only of the Taylor et al. (2015) paper. This effect has been noted by others in papers that focused on this issue (Willis et al., 2014) and others have used this knowledge to account for the variability in the detection efficiency (e.g., Collier et al., 2018). (I'll also note that Taylor et al. show the SP-AMS/SP2 BC ratio versus the absolute concentration of inorganic species measured by the SP-AMS, not versus the coating-to-BC ratio as implied by the authors use of the term "mixing state" here.) I suggest the authors temper the statement here a bit to indicate that quantification is challenging and care must be taken to account for changes in detection efficiency that might occur as the coating state of the BC changes. Also, it would be useful if the authors would clarify whether they are using "absolute" here to mean the absolute concentrations or the absolute coating

thickness. I believe they mean the latter, based on the discussion in the next paragraph. However, this seems to contradict, somewhat, the authors statement that the SP-AMS does well with the coating-to-core ratio, which is the primary determinant of the coating thickness (if one translates from a mass ratio to a coating amount).

We thank the reviewer for these suggestions and apologize for omitting the relevant references in our initial submission. We have tempered the original statement as suggested by the reviewer, added the suggested references, and clarified that we are referring to absolute mass concentrations of BC cores and their coating material. The revised statement now says (L95): "However, the detection efficiency of BC cores in an SP-AMS is sensitive to  BC mixing state, since coatings affect the focusing of BC-containing particles within the instrument, and consequently the degree of overlap between the instrument's laser and particle beams  (Taylor et al., 2015; Willis et al., 2014). This complicates the quantification of absolute BC core and coating mass concentrations with the SP-AMS, particularly in ambient campaigns where a wide variety of BC mixing states might be encountered (e.g. Collier et al., 2018)."

L87: Given the citation of the Taylor et al. (2015) paper above, it seems appropriate that the authors here might acknowledge some of the challenges in extracting absolute measures of the coating thickness that result from having to make particular assumptions regarding the BC material properties. (This does come up later, briefly.) Additionally, the SP2 interpretation method inherently assumes spheres yet the particles may not be spherical (as the authors note above), which will affect the accuracy of the method; it seems this should be noted. Also, the authors cite here Laborde et al. (2012a) as justification for "quantitative" coating thickness determination. It is unclear where in that paper coating thickness determination is shown to be quantitative; the paper seems to actually be quite qualitative in terms of coatings. As such, I suggest this reference be clarified, removed, or replaced with a more appropriate reference.

We agree this is a good place to raise this point, especially given its importance. We have modified the statement to make this point and then direct the reader to Sect. 2.4.1, where a detailed discussion of the required assumptions is already provided (we would argue this discussion can't be classified as 'brief', as is also seemingly acknowledged later by the reviewer in their comment on 'SP2 coating determination'). Regarding the Laborde et al., (2012a) citation, it is included here not for justification purposes, but rather as a reference for how such coating thickness calculations can be performed (i.e., for the same reason Gao et al., 2007 is cited). In particular, Section 2.2.1 from Laborde et al. (2012a) details an important modification to the original Gao et al. (2007) procedure for determining coating thicknesses that requires less assumptions on the laser beam profile. That discussion covers some of the key quantitative aspects in determining the scattering cross section required for the coating thickness determination. Therefore, we believe the reference is appropriate and choose to leave it in place.

The modified statements now read (L105): "In addition, incandescence measurements are combined with optical measurements of particle size in the SP2, allowing quantitative measurement of the BC coating thickness under the assumptions of a core-shell morphology for BC-containing particles as well as certain material properties (Gao et al., 2007; Laborde et al., 2012a). The derived coating thickness values are sensitive to these assumptions as discussed in detail below in Sect. 2.4.1 and by Taylor et al. (2015), which necessitates the exercise of considerable care when using the SP2 to perform quantitative coating thickness measurements".

L115: Is it quite correct to state that the MAC is compared to the "BC particle mixing state?" Coating thickness is not exactly mixing state. At minimum, the authors should clarify that they mean "internal mixing state" per their discussion earlier. But I suggest that rather than using "mixing state" here and "infer[ring]" mixing state from coating thickness they just state that they compare with mean coating thickness for particles in a particular size range from the SP2. I feel that the use of the term "mixing state" here serves to confuse rather than clarify, at least for me. The same I find true when the authors state that they determine "mixing state" by thermodenuding. I suggest just stating what specifically was measured, for example (L120) that the authors aim to infer a causal relationship between the lensing effect and the "coating amount" rather than "mixing state."

Although we agree that 'mixing state' is a broader concept then 'coating thickness' alone, we believe that our use of the term 'mixing state' is generally consistent with the relevant literature. For example, in a recent, comprehensive review of aerosol mixing state (Riemer et al., 2019), SP2 measurements of coating amounts are discussed at length with respect to previous studies in a section titled "Other Mixing State Metrics for Measurements and Models" (Sect. 5.3). Therefore, we consider it as accepted usage to refer to such measurements as a metric of BC mixing state. To clarify this point we have added reference to the aforementioned mixing state review on L142: "The relationships between this specific metric of BC mixing state and more general measures of aerosol mixing state (e.g. the mixing state index $\chi$) are discussed in the review of Riemer et al. (2019)".

Riemer, N., Ault, A. P., West, M., Craig, R. L. and Curtis, J. H.: Aerosol Mixing State: Measurements, Modeling, and Impacts, Reviews of Geophysics, 57(2), 187–249, doi:10.1029/2018RG000615, 2019.

L192: The authors note that the SP2 "missing mass correction" will be detailed in a separate paper that is not available for review at this time. Therefore, I suggest that the authors provide at least a short summary of the correction method. For example, was a single campaign average value applied, or did the authors determine a missing mass correction based on three hour averages (as this is their averaging time for the various measurements)? If the latter, noting min/max values (as percents) would seem appropriate. It seems to me that the latter is more appropriate. Certainly, based on the mean values given the missing mass correction details should have small effect, but they should nonetheless be noted here.

Originally a single missing mass percentage was calculated based on the campaign averaged size distribution. Since this value was very low a correction factor was not applied to account for the missing mass. This was not stated clearly in the original manuscript and indeed the opposite was even implied. We apologize for this mistake and the confusion it caused.

As insightfully noted by the Reviewer in their comment on line 562, it is important to consider potential differences in missing mass between the four main periods highlighted during the campaign (described in Sect. 3.1), in order to be able to examine to what extent these differences might contribute to the trends seen in e.g. Fig. 6. Therefore, we have now calculated missing mass percentages for each of the four main periods identified during the campaign. Further, the reviewer also noted in their comment on L562 that a single fitted lognormal mode might potentially underestimate the fraction of missed mass.

We agree that this is a possibility. Therefore, we fitted lognormal modes separately to the upper and lower portions of the mass size distributions to produce more conservative estimates of the missed mass percentages.

We have added a new paragraph at L234 to explicitly discuss each of these points and to explain the full process more clearly (note the cited paper by Pileci et al. has since been made available online in AMTD):

"As discussed by Pileci et al. (2020), there are a number of different methods for quantifying and correcting for the mass of BC outside the SP2 size detection limits (if the user decides to apply a correction at all). These methods are based on extrapolation of SP2-measured BC core mass size distributions. In this study we used the lognormal fit approach. Further, to better represent the upper portion of the size distributions where most of the missing mass appeared to lie (Fig. S5), we fitted lognormal functions separately to the lower ($80 < D_{rBC} < 300$ nm) and upper portions ($230 < D_{rBC} < 600$ nm) of the measured size distributions. The extrapolated portions of these two types of fits are displayed in Fig. S5 for each of the four main periods of the campaign (which are introduced and described in Sect. 3.1). From the extrapolated sections of the fitted curves we estimate the missing mass percentages below the lower LOQ were 1.1, 0.4, 1.6, and 0.8 % for periods 1, 2, 3, and the short plume case, respectively. The corresponding percentages for the missing mass above the upper LOQ were 1.3, 4.9, 4.0, and 26%. Since these estimated percentages are low (less than 5% excepting the small portion of the dataset represented by the short plume case), we chose not to apply correction factors to account for the BC mass potentially missed by the SP2. The possible consequences of this decision are discussed in Sect. 3.3."

We have also replaced Fig. S5 with a new version that displays the extrapolated portions of the lognormal fitted lines that were used to calculate the missing mass percentages. The caption of this figure has also been updated accordingly.

New version of Fig. S5:

[Figure]

SP2 coating determination: While the details provided here are most certainly important, the authors could probably move most of these details to the supplemental and then note more succinctly in the main text the method used and the key uncertainties.

Although it is a lengthy discussion it was a deliberate decision on our part to include these details in the main text. Our reasoning is that we believe these details are often overlooked in SP2 studies and that, consequently, the complexity of this analysis and the sensitivity of derived coating thicknesses to the various underlying assumptions is often overlooked. We believe it is very important to highlight this complexity and sensitivity, as also suggested by the reviewer in their comment on L87 above. For this reason we would like to keep these details in the main text rather than moving them to the supplemental.

L211: It is unclear to me how Fig. S1 shows that optical diameters are within "a few percent" of the mobility diameters. Are the authors making this statement based on some general similarity in shape? Also, in Fig. S1 I find it unclear whether the red curve is the sum of the blue and the black, as it should be based on the definitions provided. If it is the sum, this means that the entirety of the blue curve is not shown. The authors might consider visual ways to clarify.

The 'few percent' stated in the original submission was a mistake on our part. We thank the reviewer for identifying this. As can be seen in Fig. S1 the agreement between the optical and mobility size distributions is clearly not within a few percent. We have recalculated this and find the two distributions generally agree to within 20%, except for the case shown from the end of the campaign with low aerosol loadings where the agreement was within 60%.

We have modified the statement so that it accurately reflects the actual level of agreement. L268 now states: "The refractive index of the particles is assumed to be 1.50+0i, which typically provides optical size distributions that agree within 20 % with corresponding mobility size distributions, excepting some outlying cases when the total aerosol load was very low (Fig. S1)".

The reviewer is also correct that the red curves are the sum of the corresponding blue and the black curves. The hidden portions of the blue curves also caused confusion for Reviewer 1. To clarify this visually as suggested, we have replaced Fig. S1 with a version where the distributions are semi-transparent so that the entire BC-free size (blue) size distributions can now be seen. In addition, we modified the legend so that it is explicitly stated that the 'all particles' distributions are the sum of the corresponding 'BC-free' and 'BC-containing' distributions.

New version of Fig. S1:

[Figure]

The minimum detectable coating thickness seems to come from Fig. S3b. I suggest that the authors report the +/- 10% and the 10th and 90th percentile bounds as diameter equivalent. I think the +/- 10% line is nominally 10 nm and the 10th/90th percentile is 30 nm.

We're not sure if we have understood this comment correctly, but It is a challenge to report these bounds in diameter units as they will then be a function of size. For example, yes the +/- 10% line corresponds to +/- 10 nm at an equivalent diameter of 100 nm on the 1:1 line, but it is +/- 20 nm at an equivalent diameter of 200 nm, and so on. The same goes for 10/90th percentile bounds, even though these are different in that they are calculated statistics. The primary purpose of the plot is to be able to compare the calculated statistics for each period (medians and 10/90th percentiles) with the 1:1 line and associated +/- 10% bounds. We believe the plot manages to achieve this so elect to leave it as is.

L305: A minor issue, but the authors might clarify whether the AAE values here are the concurrently measured 3-h averages.

Section 2.4.2.2 has been rewritten in response to several comments:
"The absorption coefficient was quantified with a combination of PAX, MAAP and AE33 data. The absorption coefficient measured by the MAAP at λ=637 nm was adjusted to λ=870 nm, using the 3h-averaged AAE data obtained from the AE33:

$$b_{ap,MAAP,870nm} = b_{ap,MAAP,637nm} \times (637/870)^{AAE(637,870)}$$
(7)

The correlation between wavelength-adjusted MAAP data and PAX data is excellent in the range above ~1 Mm$^{-1}$ (Fig. 2). However, the $b_{ap,PAX,870nm}$ measurements are biased systematically lower than the $b_{ap,MAAP,870nm}$ measurements by 31%, which is most likely a result of imprecise and hence inaccurate PAX calibration (Section 2.4.2.1). The MAAP demonstrated good accuracy in several intercomparison studies and it has been shown to have a low instrument-to-instrument variability of less than 5 % (Müller et al., 2011). Therefore, we decided to scale the PAX data by a constant scaling factor of 1.44 [=1/(1-0.31)] to match $b_{ap,MAAP,870nm}$ as shown in Fig. 2. Application of this constant scaling factor brings the scaled PAX and MAAP measurements into good agreement at absorption coefficients greater than ~15 Mm$^{-1}$, while the scaled PAX measurements are up to 10% lower than the corresponding MAAP measurements in the range down to 1 Mm$^{-1}$ (as shown by the green crosses in Fig. 2). The scaled PAX data provide absorption coefficients with high time resolution and for both ambient and denuded inlets, with absolute calibration referenced to the MAAP."

L308: It would be helpful if the authors clarify how a 31% bias translates to a scaling factor of 1.44. Why not 1.31?

See new Section 2.4.2.2 provide in response to the previous comment.

Fig. 2: Visually, it appears that the difference between the 1:1 line and the scaled measurements is largest at intermediate absorption values. It would be useful if the authors were to consider the percent difference as a function of absolute absorption, and comment on how this might (or might not) impact their conclusions here.

Thanks for this suggestion. We have replaced Fig. 2 with a version that displays the residual percentages between the scaled PAX and MAAP measurements. The caption of Fig. 2 has been modified accordingly and on L382 we have added the following sentence: "Application of this constant scaling factor brings the scaled PAX and MAAP measurements into good agreement at absorption coefficients greater than ~15 Mm$^{-1}$, while the scaled PAX measurements are up to 10% lower than the corresponding MAAP measurements in the range down to 1 Mm$^{-1}$ (as shown by the green crosses in Fig. 2)."

To consider the potential impact of a loading-dependent scaling factor on our conclusions we have added a new supplementary figure to the manuscript, Fig. S10. Panel a) of this figure is equivalent to Fig. 6a, panel b) displays the same quantities but with an additional correction factor of 1.1 applied to the MAC of BC values corresponding to absorption coefficients less than 15 Mm$^{-1}$, panel c) displays the same quantities but with period-dependent missing mass correction factors applied to the MAC of BC values (as suggested by the reviewer in their comment on L562), and panel d) displays the same quantities but with both the loading-dependent absorption scaling factors and period-dependent missing mass correction factors applied to the MAC of BC values. It is seen that although the spread in the data points increases when the different types of scaling factors are applied to the MAC of BC data, the clear positive relationship between the MAC of BC and mean coating thickness remains. Therefore, these changes have no impact on the main conclusions of the manuscript.

We have added the following sentences on L598 to introduce Fig. S10 and discuss these implications: "To investigate the robustness of the results of the correlation method, Fig. S10 displays versions of Fig. 6a with different scaling factors applied to the underlying quantities used to calculate MAC$_{BC}$. In particular, we investigated the effect of applying a loading-dependent absorption scaling factor in Fig. S10b (as motivated by Fig. 2 and the discussion in Sect. 2.4.2.2), as well as the effect of applying separate missing rBC mass correction factors for each time period of the campaign in Fig. S10c (as discussed in Sect. 2.4.1). In these cases (as well as the case when both types of scaling factors are applied, Fig. S10d), the positive correlation between MACBC and the mean coating thickness remains, supporting the evidence for an observed lensing effect."

L322: Since the losses are size dependent, and since coating-to-BC ratios are likely size dependent as is the SP2 detection efficiency, the authors might instead state that the losses likely introduced minimal bias, rather than stating categorically that they introduced no bias.

We agree it cannot be categorically proven that no bias was introduced. We have tempered the statement as suggested. L414 now states: "The losses likely introduced only negligible bias in the MAC$_{BC}$ values since the absorption measurement by the PAX was also behind the CS."

L323: The shapes of the particles may have changed upon denuding. Might this impact the determination of coating thickness, as the method assumes spherical particles (even for the BC)?

Coating thickness was exclusively quantified by SP2 data. Denuded data were only used to obtain an upper limit for the MAC$_{BC}$ of the bare BC core. The MAC$_{BC}$ of the bare BC core could be slightly affected by denuding compaction (which could potentially add up to the condensation compaction that already occurred during coating acquisition). However, the systematic bias introduced by residual coatings after denuding is likely larger and this is accounted for by using the denuding and the correlation approaches to obtain upper and lower limits of the MAC$_{BC}$ of bare BC cores (and respective lower and upper limits of the absorption enhancement factor). We added the following statement at the end of Sect. 2.4.3: "Denuding could potentially cause some compaction of the BC cores. However, the denuded sample data were only used to determine the MAC$_{BC}$ of the bare BC cores – more precisely an upper limit of it

due to residual coating – and hence such compaction does not significantly affect the interpretation of our results."

L416: It is not clear to me how Fig. 2 indicates the lower limit of quantification for the PAX. It just shows the relationship between the PAX and MAAP. The limit of quantification typically comes from consideration of the instrument noise when sampling zero particles over the time period of the instrument zeros.

Agreed. The value of ~1 Mm$^{-1}$ is the value quoted by the manufacturer. Our intention was to indicate that Fig. 2 is at least consistent with that. We have modified the sentence on L517 as follows: "(the manufacturer-reported sensitivity of the PAX is <1 Mm$^{-1}$ at 60 secs averaging time, which is consistent with instrument performance demonstrated in Fig. 2)".

L438: It would be helpful if the authors could clarify why they are not considering emissions from biomass combustion, which tend to also produce larger BC cores, instead focusing on coal as the major BC source.

Our original focus on coal emissions stemmed from the discussion in the previous section (Sect. 3.1), where we demonstrated that period 2 was characterized by air mass transport from Poland where high-sulfur content coal is still burned for industrial and domestic purposes. Elevated $SO_2$ to $NO_x$ ratios were also measured during period 2, supporting the hypothesis that coal emissions were indeed an important source of BC during this period. However, we agree that biomass burning (e.g. wood) emissions might also be explain the larger BC cores observed during this period.

We have modified the discussion on L542 to explicitly mention this alternative (and not mutually exclusive) hypothesis: "The larger BC particles measured in period 2 might be related to coal burning emissions (e.g. lignite coal burning in Poland): while the burning of hard coal briquette emits particles that lie mostly in the nuclei and Aitken mode (20-100 nm), the number size distribution of lignite emissions peaks in the accumulation mode (100-1000 nm) (Bond et al., 2002). Therefore, it is possible that BC cores from lignite burning are larger than BC from other common sources such as traffic. Wood burning emissions from domestic heating are also expected to generate larger BC cores than those emitted by traffic, and thus could also be partly responsible for the generally larger BC cores observed during period 2."

L468: Do the results here change if the authors instead use the median values (rather than the mean)? Or, what if the authors use a geometric average rather than an arithmetic average. The former is typically more appropriate for distributions that are bounded on one side. Here, the coating thickness cannot be <0 nm (within error), and thus the distributions are inherently non-Gaussian but instead more log normal. (Also, Fig. S7 reports the medians. It would be good to see things reported in a common way.)

In general we agree that the median or geometric mean would be a more appropriate statistic for summarizing the coating thickness distribution itself. However, in this case our aim was to calculate an

'ensemble mean' coating thickness value that could be related to the measured absorption enhancement factors (which are ensemble averages by virtue of the way they are measured). Ideally, this would be done by weighting the calculated coating thickness average with the true coating-dependent enhancement factors. However, the true relationship between enhancement factor and coating thickness is unknown (indeed we seek to measure it). Nevertheless, our results suggest that this relationship is linear in fair approximation (Fig. 6a). Therefore, we chose to apply an arithmetic rather than geometric mean to represent the 'ensemble mean' coating thickness values.

In any case, the results do not change substantially regardless of the statistic used. To demonstrate this and to report things more consistently as suggested by the reviewer we have added the arithmetic mean values to the legends in both Figs. S7 and S8. These figures now report both the median and mean values so that the reader is able assess the difference between the two different measures.

Fig. 6: Given that the authors use the standard error of the mean to show their uncertainties here, it would be helpful if they would report the typical instrumental averaging times so that the reader can know how many points go into the 3-h averages. (In other words, are these averages of 1s data? 1 min? 10 min?)

The MAC of BC averages were calculated with 1 min resolution data, while the mean coating thicknesses were calculated with 1 sec resolution data. We have added this information to the caption of Fig. 6, although we note that the standard error of the mean is of course a meaningful quantity on its own. Knowledge of the number of points that go into its calculation is only required if one wished to interpret the standard deviation which is affected by increasing time resolution at the expense of increased random noise of underlying data points.

Fig. 6b: Given that the denuded particles appear to still have coatings, I suggest that it would be helpful to show also a composite plot with the ambient particles versus their coating thickness and the denuded particles versus their coating thickness so that the continuity between these can be seen.

We agree that some readers might find this plot useful (particularly because Reviewer 1 also asked to see it). We have added it as a new figure in the supplementary information (Fig. S4). The figure shows that there is no clear correlation between the denuded MAC_BC values and the coating thickness of the denuded particles

We have modified the statement beginning on L619 to refer to the new Fig. S4: "It is seen that most of the denuded-$MAC_{BC}$ values fall in the range from 5.6 to 6.0 $m^2$ $g^{-1}$ (median= 5.8 $m^2$ $g^{-1}$) and that the values are largely independent of the original coating thickness, with a Pearson correlation coefficient of only 0.02 (the denuded-$MAC_{BC}$ values are also largely independent of the coating thickness of the denuded particles as shown in Fig. S4, with a Pearson correlation coefficient of -0.29)."

Here is the Fig. S4:

[Figure]

L497: It is not accurate to state that "all" the denuded MAC values fall in the stated range. There are some points that are outside this range. "Most" of the denuded MAC values fall in the stated range.

Agreed. We apologize for the careless wording in the original statement (as also noted by Reviewer 1). We have modified the statement to state "…that most of the denuded-MAC$_{BC}$ values fall…" in the stated range (L619).

L549: Biomass burning emissions also exhibit a wide range of AAE values. Thus, it is not clear why apportionment is appropriate in the biomass burning assumption above but not for coal. It is also not clear why the biomass burning method would provide an "upper limit estimate" for the contributions. How is it specifically known that this is an upper limit?

The difference is that in the case of biomass burning previous studies indicated that the AAE of the typical central European mix of biomass burning aerosol is reasonably constant (e.g. Zotter et al., 2017). To the best of our knowledge no such studies have been performed for central European coal emissions as BC from biomass burning dominates over BC from coal burning in most areas. We have modified the sentence on L676 to include this explanation: "Given the wide range of observed AAE for coal burning emissions, and the lack of knowledge regarding a specific value that is appropriate for central Europe (e.g. as is the case for biomass burning aerosols; Zotter et al., 2017), it is not possible to use measured AAE values to apportion a specific fraction of equivalent BC to coal emissions."

Regarding the upper limit, we suggest these values are upper limits since if a third source was included in the aethalometer model than the contributions from the original two sources would most likely be lower. We have added this explanation to the sentence of L684: "In this case, the contributions of wood burning and traffic to BC as calculated with the aethalometer model and displayed in Fig. S9b should be considered as upper limit estimates, since these contributions would likely be lower if a third source was included in the model."

L 562: I suggest it would be useful for the authors to calculate and report the R2 value when the two clear outlier points are excluded. These correspond to the short period when the BC source was quite different. The Rˆ2 in this case would definitely be >0.01, although still not as large as the relationship with the coating thickness shown previously. Also, I find it a little awkward to note that variability in the BC diameter over the range 150 nm to 250 nm is a "narrow range" but the coating thicknesses, that only varied from _40-60 nm is not "narrow." Also, visually there appears some correlation between the BC diameter and the coating thickness from Fig. 5 (excluding the short plume). Finally, from the size distributions shown in Fig. S5, it would seem that any missing mass correction would be larger for period 2 than for period 3 than for period 1. To what extent might uncertainty in the missing mass correction contribute to the relationships shown? The distributions are close to log-normal, but not quite and thus use of a single mode fit might underestimate the correction that is necessary.

Taking each of the questions in this comment in turn:

The R^2 value with the two outlying points removed is 0.09. The corresponding Pearson correlation coefficient is 0.31. As predicted by the reviewer this is well below the correlation coefficient for the MAC of BC relationship against coating thickness (Fig. 6; 0.79). In addition, the relatively even spread of data points across the full range of measured coating thicknesses lends greater confidence to the latter measurement of correlation. In contrast, the clustering of the data points into two main clouds when plotted against the modal diameter of rBC indicates that the calculated correlation coefficient of 0.31 should be interpreted with caution. Therefore, removing the two outliers has no effect on the conclusions drawn. Nevertheless, we have added the additional measure of R^2 to the parentheses on L694: ". Figure 7b shows that there was no clear relationship between $MAC_{BC}$ and BC core diameter ($R^2$=0.01, or 0.09 with the two outlying points with the largest modal diameters removed), which indicates that the variation in BC size was not responsible for the $MAC_{BC}$ variability."

Regarding the use of the word narrow, we are not sure at which part of the manuscript the reviewer is referring where the range of measured coating thickness values is described as "not narrow". In any case, we would argue such a description could indeed be appropriate, depending on the context. For example, the range of coating thicknesses measured during this study are broad relative to those that have been measured in other studies using the same technique (e.g. this can be seen by comparing the ambient data shown in Fig. S8 with the compilation of coating thickness histograms presented in Fig. S6 of Motos et al., 2020). In contrast, we consider the rBC core diameter range from 150 to 250 nm to be "relatively narrow" in the context of the discussion that immediately follows on from this sentence. To summarize this discussion, this diameter range corresponds to size parameters from 0.5 to 0.9 at a wavelength of 870 nm, which is not wide enough to extend into the Rayleigh regime at the lower end and the geometric regime at the upper end. In this context, which is already outlined in the paragraph in question, we believe the use of the phrase "relatively narrow" is appropriate.

As described in the response to comment on L192 above, the potential missing mass correction is indeed greater in period 2 than in period 1 (though not greater than in period 3, considering also the potentially missed mass below the lower detection limit). As described above since these missing mass corrections are small (excepting the short plume case) and can't be known with any certainty (since they are calculated from extrapolated curves) we elected not to apply them. To consider the potential impact on the results if these period-dependent correction factors were applied, we added a new supplementary figure (Fig. S10) to the manuscript as described in our response to the comment on Fig. 2 above. This figure shows that although the spread in the data points increases when the different types of scaling factors are applied to the MAC of BC data, the clear positive relationship between the MAC of BC and mean coating thickness remains

We have added the following sentences on L598 to introduce Fig. S10 and discuss these implications: "To investigate the robustness of the results of the correlation method, Fig. S10 displays versions of Fig. 6a with different scaling factors applied to the underlying quantities used to calculate MACBC. In particular, we investigated the effect of applying a loading-dependent absorption scaling factor in Fig. S10b (as motivated by Fig. 2 and the discussion in Sect. 2.4.2.2), as well as the effect of applying separate missing rBC mass correction factors for each time period of the campaign in Fig. S10c (as discussed in Sect. 2.4.1). In these cases (as well as the case when both types of scaling factors are applied, Fig. S10d), the positive correlation between MACBC and the mean coating thickness remains, supporting the evidence for an observed lensing effect."

Motos, G., Corbin, J. C., Schmale, J., Modini, R. L., Bertò, M., Kupiszewski, P., Baltensperger, U. and Gysel-Beer, M.: Black Carbon Aerosols in the Lower Free Troposphere are Heavily Coated in Summer but Largely Uncoated in Winter at Jungfraujoch in the Swiss Alps, Geophysical Research Letters, 47(14), e2020GL088011, doi:10.1029/2020GL088011, 2020.

L567: The authors might note that this is the theoretical result of Mie theory, which might not be fully applicable to fractal-like particles for which absorption by the individual spherules might dominate.

We disagree with this comment. The inverse relationship between MAC of BC and BC diameter for size parameters much greater than 1 is not strictly a result of Mie theory. For example, it also emerges from geometric optics (e.g. Moosmüller and Sorensen, 2018). In addition, the inverse relationship is also observed in numerical computations of the MAC of BC for soot aggregates (Fuller et al., 1999; Mackowski, 1994). Therefore, we think it is appropriate to leave this general statement as is.

Fuller, K. A., Malm, W. C. and Kreidenweis, S. M.: Effects of mixing on extinction by carbonaceous particles, Journal of Geophysical Research: Atmospheres, 104(D13), 15941–15954, doi:10.1029/1998JD100069, 1999.

Mackowski, D. W.: Calculation of total cross sections of multiple-sphere clusters, J. Opt. Soc. Am. A, JOSAA, 11(11), 2851–2861, doi:10.1364/JOSAA.11.002851, 1994.

Moosmüller, H. and Sorensen, C. M.: Small and large particle limits of single scattering albedo for homogeneous, spherical particles, Journal of Quantitative Spectroscopy and Radiative Transfer, 204, 250–255, doi:10.1016/j.jqsrt.2017.09.029, 2018.

L578: I would suggest the authors go a bit further and argue that it is very likely that the coating amounts differed, as no attempt was made in that study to ensure that the particles had the same amount of coating.

Agreed. We have modified this sentence slightly to reflect this (L716): "However, it should be noted that the amount of coating could not be quantified conclusively in these experiments, and therefore it must be considered likely that the coating volume fractions differed for the different coating compositions."

Fig. 8 and origin of Ueda et al. (2016) data: The authors show results from Ueda et al. (2016) in this figure. In their caption they note that the points shown might differ from those shown in Cappa et al. (2019). However, in looking at Cappa et al. (2019) the Ueda et al. data are not included. As such, it is unclear where these data come from. Looking at the Ueda et al. (2016) paper directly, there is no indication that values of R_coat-BC are available in a general sense. At best, it would seem that up to four data points might be included, corresponding to the periods A-D in Ueda et al. (2016). It looks like the authors used the mean volume fraction of soot for particles in varying size ranges and with varying number of particles analyzed (per Ueda et al., Table 3) to calculate the volume fractions. For one of these periods (C), only 6 particles were analyzed, calling into question the statistical significance of any volume fraction. As for the Eabs values, did the authors use the 400 degC results reported? Or did they calculate an estimated Eabs based on the reported [BC] and absorption at 781 nm? I suspect the former, but it might be noted that if one assumes a constant MAC for uncoated BC and calculates an Eabs from the reported measurements, the thus derived Eabs does not match with the value derived from heating to 400 degC in terms of their apparent dependence on the coating volume fraction.

We agree with these concerns about the Ueda et al., data as represented in this plot. The reviewer is correct that for one of the data points only 6 BC particles (with diameters < 600 nm) are included in the calculation. As suggested, this makes it difficult to determine a representative BC volume fraction that corresponds to the overall absorption. For the Eabs calculation, the 400 degC results at 781 nm were indeed used (Table 2 of Ueda et al.), and we agree that the elevated temperature needs to be taken into consideration. Including all these caveats in the description of Fig. 8 obscures the overall message of the figure without adding any additional insight. Therefore, we decided to remove the Ueda et al., data from Fig. 8 in the revised manuscript.

Fig. 8b: It is unclear whether the x-axis is really the total NRPM-to-BC ratio for all studies shown. For some of the studies, the x-axis values in panel b appear the same as in panel a, which cannot be the case unless all NRPM is internally mixed with BC. But for other studies the relationship differs. Additionally, it is also not clear for the current measurements that the x-axis is correct. Looking at Fig. 2, the BC fraction ranges from about 0.04 to 0.22, excluding the short plume. The NRPM/BC ratio should just be (1-BCfraction)/BCfraction, and so should range from approximately 3.5 to 24. But the data shown in Fig. 8b only range from 2-4 for the current study. I suggest that clarification is needed. Perhaps I am just

misunderstanding the relationship between the R_coat-BC and M_total/M_BC as used here. Or misunderstanding what the authors mean in the figure caption when they note that the M_total/M_BC is a "free parameter." Free in what way? (As a minor note, inclusion of ticks between 1 and 2 on the y-axis would be helpful to the reader.)

We thank the reviewer for pointing out these points where clarification is required. The quantity plotted on the x-axis of Fig. 8b is the ratio of the mass of the total particle to the mass of BC in the BC-containin particle, not the NRPM or coating to BC mass ratio. This is quantity is related to $R_{coat\text{-}BC}$ (from Fig. 8a) according to: $M_{total}/M_{BC} = 1 + R_{coat\_BC}$. It was described as the 'free parameter' in the Chakrabarty and Heinson parameterization in the sense that it is the independent, input parameter to this function. We agree the use of the word 'free' can be confusing. To clarify these points we have made the following additions:

The caption of Fig. 8 now states: "Figure 8. Summary of $E_{MAC}$ values and their dependence on BC mixing state from this study compared with literature data measured at wavelengths from 532 nm to 870 nm. The $E_{MAC}$ values are plotted versus the ratio of coating to BC core mass (a) and versus the ratio of total particle mass to BC core mass (b). Panel (a) is an updated version of a figure from Cappa et al. (2019), while in panel (b) the abscissa has been changed to $M_{total}/M_{BC}$ to additionally include the simple power-law parameterization of the lensing effect proposed by Chakrabarty and Heinson (2018), which uses $M_{total}/M_{BC}$ as the input parameter to represent BC mixing state (where $M_{total}$ refers to the total mass of the BC-containing particle, such that $M_{total}/M_{BC} = 1 + R_{coat\text{-}BC}$). The two approaches applied to obtain the lower and upper limit $E_{MAC}$ values presented for this study are discussed in Sect. 3.3). Note: the data points from the study of Liu et al. (2015a) deviate marginally from those in the original figure by Cappa et al. (2019), as the abscissa values were recalculated from the original data using material densities for BC and coating material of 1.8 and 1.6 g cm$^{-3}$, respectively, as applied in this study."

And on L830 of the main text: "The $E_{MAC}$ results from this study and selected ambient literature studies are plotted against BC mixing state expressed as $M_{total}/M_{BC}$ in Fig. 8b (where $M_{total}/M_{BC} = 1 + R_{coat\text{-}BC}$)."

We have verified the correctness of the x-axis values in Fig. 8b. We believe confusion resulted from the fact that Mtotal/MBC was not adequately defined as being equal to 1 + Rcoat-BC. Due to the log x-axis, this means Rcoat-BC values greater than 10 don't seem to have moved much by eye when transformed to the Mtotal/MBC axis. The change is more noticeable, however, for the smaller Rcoat-BC values between 1 and 10, and even more so for values less than 1.

Regarding the measurements from this study specifically, we have also verified that these calculations are correct. The reviewers reasoning is essentially correct. However, confusion was caused by a mistake in Fig. 5d. The y-axis label of this figure mistakenly described the BC volume fraction as a percentage, not a fraction. Thus, the volume fractions from 0.2 to 0.4 correspond to Mtotal/MBC values of ~2 to 4 (assuming coating and BC core material densities of 1.6 and 1.8 g/cm3, respectively). We have corrected Fig. 5 so that the '%' symbol has been removed from the axis label.

Finally, we have added minor ticks between 1 and 2 on the y-axis of Fig. 8 as suggested by the reviewer.

L638: I suggest it would be helpful for the authors to be more explicit about the SP-AMS here. The introduction section does not indicate that the coating-to-core ratio from the SP-AMS should depend strongly on the coating amount but instead notes that absolute quantification of concentrations depends on the coating amount. How does a change in the collection efficiency affect the ratio, rather than the absolute values? Also, they might note that the laboratory studies that have looked at this effect find that above a coating-to-core ratio of _3 that the collection efficiency is effectively constant, and some of the studies shown do have coating-to-core ratios this large.

We agree that these are important clarifications and that our initial statement was overly simplistic. A change in collection efficiency could affect the ratio Rcoat-BC if the change in apparent efficiency was different for the non-refractory coating material and the BC core (e.g. as appears to be the case for organic coatings as shown in Fig. 2 of Willis et al., 2014, a reference mentioned by the reviewer above). Regarding the point that collection efficiency of the SP-AMS appears to be constant for Rcoat-BC values greater than 3, we believe the reviewer is referring to the same Fig. 2 of Willis et al., 2014. If so, this statement is then only based on 2-3 data points. Therefore, we do not feel comfortable generalizing this result to all the measured values greater than 3 displayed in Fig. 8a.

All in all, we feel that such detailed discussions of SP-AMS measurements are beyond the scope of this manuscript, given that the study is based on SP2, not SP-AMS measurements. We believe the most important point is that some differences are indeed observed between the two types of studies shown in Fig. 8b, which motivates the conclusion that dedicated, follow-up intercomparison studies are required. To focus on this point we have removed the sentence about SP-AMS collection efficiency. L789 now simply states: "It is noteworthy that the majority of SP-AMS-based studies observed only a weak lensing effect".

L640: This is a really great point, that I suggest the authors re-emphasize specifically in their conclusions.

As suggested we have re-emphasized this point in the conclusions section. L869: "In addition to this challenge, follow up studies should aim to intercompare different techniques for measuring BC volume fractions (e.g. SP2 versus SP-AMS measurements of $R_{coat-BC}$)."

In their discussion of physical explanations for the differences between studies, the authors might note that some of the studies are more likely to be impacted by local sources and some by particles that have undergone long-range transport. The latter might tend to homogenize the population, which could affect the observable absorption enhancement.

As suggested we have added a sentence about this on L813: "The interplay of these different effects and the resulting observable enhancement are likely to depend on the ratio of fresh to aged BC particles in a given air mass, and will therefore vary from site to site."

L681: The sentence beginning "The enhancement: : :" does not seem to be a complete sentence.

Thanks for picking this up. We have changed this sentence L847 to: "The enhancement of light absorption by BC due to coatings varied from 1.0 to 1.6 (lower limit estimates), or 1.2 to 1.9 (higher limit estimates), for mean BC volume fractions that varied from 46 to 22 % (for BC core sizes from 200 to 220nm)".

L691: It is not overly clear to me how "such effects" might be obscured here. What do the authors mean when they indicate "a more quantitative assessment?" Do they mean calculated across the entire size distribution?

We have modified this sentence to clarify that we are referring to quantitative assessment on the single particle level, L861: "However, such effects could potentially be obscured by the lack of single particle composition and morphology information that would be required for a single-particle level quantitative assessment."

References

Collier, S., Williams, L. R., Onasch, T. B., Cappa, C. D., Zhang, X., Russell, L. M., Chen, C.-L., Sanchez, K. J., Worsnop, D. R., and Zhang, Q.: Influence of emissions and aqueous processing on particles containing black carbon in a polluted urban environment: Insights from a soot particle – aerosol mass spectrometer, Journal of Geophysical Research-Atmospheres, 123, 6648-6666, https://doi.org/10.1002/2017JD027851, 2018.

Taylor, J. W., Allan, J. D., Liu, D., Flynn, M., Weber, R., Zhang, X., Lefer, B. L., Grossberg, N., Flynn, J., and Coe, H.: Assessment of the sensitivity of core / shell parameters derived using the single-particle soot photometer to density and refractive index, Atmos. Meas. Tech., 8, 1701-1718, https://doi.org/10.5194/amt-8-1701-2015, 2015.

Willis, M. D., Lee, A. K. Y., Onasch, T. B., Fortner, E. C., Williams, L. R., Lambe, A. T., Worsnop, D. R., and Abbatt, J. P. D.: Collection efficiency of the soot-particle aerosol mass spectrometer (SP-AMS) for internally mixed particulate black carbon, Atmospheric Measurement Techniques, 7, 4507-4516, https://doi.org/10.5194/amt-7- 4507-2014, 2014.

---

## Author Response (AR2)

Dear Editor Kanaya and Reviewer #1,

Thank you for the careful evaluation of our revised manuscript. We have implemented the suggested technical checks as described below. Editor and reviewer comments are in black text and our responses are given in blue text.

1. If not separately defined, notation of "ambient MAC_BC" in line 19, "MAC_BC,amb" in lines 20, 741 and 743, "MAC_BC,mixed" in lines 59 and 60 (equation (2)), MAC_BC,ambient in equation (4) can be unified? MAC_BC,denuded and MAC_BC,bare should be different due to the coating remaining after heating but their use in lines 20, 60, 148, 564, 586, 591, 593, 595, 605, 606, 715, etc. would be better rechecked.

We have unified the notation for the ambient MAC of BC so that all instances are described as "$MAC_{BC,ambient}$". In addition, we added a small clarification before Eq. 4 on line 147 to explicitly indicate that within the denuding method we assume that the denuded MAC of BC represents the bare MAC of BC. Specifically, we added the following statement: "(i.e., by assuming that $MAC_{BC,denuded} = MAC_{BC,bare}$)".

Figure S1: All gray color of BS-containing particles is looked the other colors by overlap with all particles. I suggest to use such as mesh rather than semi-transparent color.

We tried using different types of meshes for this figure rather than semi-transparent color but it did not work well. Instead we decided to retain the semi-transparent shading but to add different types of line styles to the envelopes. We believe this is the best solution for being able to visually identify the three overlapping distribution.

Further modifications:

Now that the manuscript is accepted for publication we have also updated the 'Data availability' statement. This now states: "The data archive for this manuscript is available on Zenodo (http://doi.org/10.5281/zenodo.4290328)"